statistical physics/biomathematics

irrationality, betting strategies, risk preferences, optimal stopping

**Author for correspondence:**
Feng Fu
e-mail: fufeng@gmail.com

# Understanding gambling behaviour and risk attitudes using cryptocurrency-based casino blockchain data

Jonathan Meng[1] and Feng Fu[1,2]

[1]Department of Mathematics, Dartmouth College, Hanover, NH 03755, USA
[2]Department of Biomedical Data Science, Geisel School of Medicine at Dartmouth, Lebanon, NH 03756, USA

FF, 0000-0001-8252-1990

The statistical concept of gambler's ruin suggests that gambling has a large amount of risk. Nevertheless, gambling at casinos and gambling on the Internet are both hugely popular activities. In recent years, both prospect theory and laboratory-controlled experiments have been used to improve our understanding of risk attitudes associated with gambling. Despite theoretical progress, collecting real-life gambling data, which is essential to validate predictions and experimental findings, remains a challenge. To address this issue, we collect publicly available betting data from a *DApp* (decentralized application) on the Ethereum blockchain, which instantly publishes the outcome of every single bet (consisting of each bet's timestamp, wager, probability of winning, userID and profit). This online casino is a simple dice game that allows gamblers to tune their own winning probabilities. Thus the dataset is well suited for studying gambling strategies and the complex dynamic of risk attitudes involved in betting decisions. We analyse the dataset through the lens of current probability-theoretic models and discover empirical examples of gambling systems. Our results shed light on understanding the role of risk preferences in human financial behaviour and decision-makings beyond gambling.

## 1. Introduction

The client server model, the most widely used computing network model in the world today, allows devices (clients) to request services or resources from other devices (servers). The client initiates a request to the server and receives a response, which usually gives the client the service or resource it requested. Some examples of this are the World Wide Web, or email. A major issue with this model is that if the server stops

working, everything else also ceases functioning. Additionally, if hackers manage to break into the server, they could steal any client information (e.g. Social Security numbers, Credit Card information) stored inside. This model inherently leads to centralization of computing power towards larger entities, such as government or multinational corporations [1].

By contrast, a peer-to-peer network lets any of its members (nodes) share information or services on the network. All nodes have equal privilege, which means any node in the network can give another node in the network a desired resource or service. The most famous example of peer-to-peer networking is in torrenting, where an initial server, called a seed, uploads a file. Nodes of the torrent network (the swarm) divide up this file into pieces and request missing pieces from other computers in the network. Once pieces are obtained by a client, or downloading node, the pieces are constructed into the original file. In this way, computing power is not monopolized; it is shared [2]. This model is both fault-tolerant (i.e. continues to work even if a single or multiple members fail) and decentralized.

Ethereum is a distributed, peer-to-peer computing network, released in 2015, that allows its nodes to conduct transactions and build applications [3]. On the Ethereum network, the main currency, Ether, powers all peer-to-peer transactions for goods and services [4].

One of the most important features of Ethereum is its usage of blockchain technology. The blockchain is a decentralized, publicly available chain of transactions. Anyone can download software (Geth, Parity) and turn their computer into a node, or a member of the Ethereum network. The peer-to-peer nature of the network allows computing power to be evenly distributed and accessible. Because all nodes contain a copy of the blockchain, each node has access to the same information. All nodes retain perfect information and verify transactions. Through the usage of blockchain technology, Ethereum aims to shift the current paradigm of computing from the client-server model to a decentralized, peer-to-peer model.

All nodes verify transactions in order to ensure that new transactions are not fraudulent. Once enough transactions are verified, these transactions are packaged together into a block. Certain nodes, called miners, then compete to compute a difficult cryptographic hashing problem, called ETHhash. This system, which rewards miners for work done is referred to as a Proof of Work System. Once a miner solves the problem, the mined block is then added to the blockchain.

After successfully packaging a block, miners are awarded with currency that is used to pay for transactions, such as Ether, or Bitcoin. On the Ethereum network, this reward is up to 5 Ether. Because each transaction is verified by all the nodes in the network, blockchains are extremely resistant to attempts of fraudulent modification. If an attacker attempts to change the system, they would have to generate an alternative chain from scratch. According to the original white paper (specification) of Bitcoin, the block synchronization of these two parties is modelled as a binomial random walk. From this, we see that the effective probability of an attacker succeeding in creating a fraudulent blockchain approaches 0 if the attacker is more than 25 blocks behind the actual blockchain [5].

Another important feature of blockchain technology is that it allows user-to-user transactions to be psuedoanonymous. This is due to a hashing of the transaction IDs and their corresponding wallet IDs. This is extremely important, as it allows for transparency of data [6]. Users do not have to worry about exposing their identity to the public. In recent years, publicly available blockchain data has attracted growing interest from diverse fields to explore human behaviour and in particular online transactions in a wide range of blockchain-based applications [7–11].

Ethereum has also introduced the idea of programming blockchain operations through a technology called the smart contract. A smart contract is an automated script written in Ethereum's own scripting language, Solidity, that allows an individual to exchange a specified good or service. A popular comparison for smart contracts is the vending machine. If a user of the smart contract gives the vending machine a certain fee, a product comes out. Accordingly, if a user inputs some cryptocurrency into a smart contract, it executes an exchange of goods or services. As smart contracts are also automated, they erase the need for a middleman. Smart contracts, if programmed properly, can be used for a variety of applications, such as vote automation or tax collection. Building an application on top of a smart contract creates a decentralized application (DApp). A DApp is completely decentralized (no single owner) and automated by its associated smart contract. Currently, there are around 1539 DApps on the Ethereum blockchain [12].

We study the behavioural dynamics of gamblers on a DApp known as Etheroll. Etheroll simulates a virtual dice gambling game where all bets are made in Ether and published on the Ethereum blockchain. Etheroll has an associated smart contract on the Ethereum network which specifies house edges, payouts and dividends to investors [13]. To begin the dice game, the gambler chooses a number between 2 and 99 (inclusive). The probability that the gambler wins is the number he or she chooses, minus 1, meaning that

the gambler can choose between a 1% to 98% chance of winning. The payout ($P'$) formula, if the house commission per bet is $e = 1\%$, probability of winning is $p$, and initial wager is $W$, is

$$P' = W\left(\frac{1 - p - e}{p}\right).$$

The smart contract then simulates a hundred-sided dice roll. If the result of the dice roll is any number smaller than the number the gambler chose, the gambler wins. After the transaction between the smart contract and the gambler processes, the gambler receives a payout (in Ether) directly to their Ethereum wallet which is inversely proportional to the probability they bet at. Naturally, lower probabilities of winning have higher payouts, and higher probabilities of winning have lower payouts. Regardless of their chosen winning probabilities $p$, the expected payout of gamblers is negative, $E[P'] = -eW$, due to the house commission fee charging $e$ percentage for each bet $W$.

These transactions are publicly available on the Ethereum blockchain. Due to the massive amount of verifying nodes on the Ethereum network, we can be sure about the validity of these transactions. We will explore these data for all four of Etheroll's smart contract updates from 17 April 2017 to 12 December 2017. Obtaining real-life gambling data, especially data from gambles in casinos is very difficult, if not impossible to obtain. Because of this, mathematical models pertaining to gambling are almost entirely theoretically based. Every bet from Etheroll consists of the bet's timestamp, wager, probability of winning, userID and profit. With these data, we shall empirically explore gambling behaviour and risk attitudes in light of the cumulative prospect theory [14].

This dataset has many other interesting properties. Having access to timestamps allows us to identify possible changes in strategy influenced by gambling results over time, in their gambling patterns. The fact that gamblers are able to tune their own betting probabilities is also crucial. The ability to tune the effective odds in a wager allows us to evaluate probable risk profiles of certain gamblers. Additionally, we focus on characterizing the entire risk attitudes of the entire gambling ecosystem as a whole. We are also able to evaluate the existence and usage of staking gambling systems (path-dependent strategies). The unique completeness and continuity of these data also allows to us empirically evaluate some famous psychological frameworks, such as the cumulative prospect theory [14]. We also characterize and quantify the effect of a gambler's cumulative 'signal', or scaled cumulative profit on their winning probability distributions and betting strategies. This scaling allows us to model the lessened effect of losses and gains over time.

# 2. Results

## 2.1. Overview

This population of gamblers on the Ethereum blockchain allows us to empirically observe the tendencies of gamblers in a casino-like environment for the first time. The minimum bet-sizing of 0.1 Ether (4–53 USD in this dataset) simulates casino-like stakes [15]. The game these gamblers play is simple, parametrized only by the probability of winning they chose and their wager size. We will first characterize the types of gamblers in this online casino through these two variables, the overall distribution of these two variables, and the paired cohort of winning gamblers and losing gamblers. We will also look at how gamblers behave when conditioned on the previous bet. To do this, we will measure the absolute and relative changes in both the probabilities they chose and their wager sizing. We will also measure the cumulative profit of gamblers—to understand how many gamblers actually end up winning anything. Lastly, we will look at gambling strategies and gamblers of interest.

## 2.2. Wager sizing

To first characterize this population of gamblers, we visualize the total bet frequency distributions of each gambler. Using histograms, we track each gambler's total gambles per smart contract, and the corresponding frequency of occurrence. In doing so, there is a very pronounced right skew in the distribution of the amount of gambles of each gambler. In fact, in each smart contract iteration, the gamblers who gamble only 1–10 times comprise approximately 60–65% of the entire population. This heavy right skew shows that most gamblers in this DApp are mainly recreational gamblers who place anywhere from 1 to 10 bets (see table 1 and figure 1).

An interesting qualitative feature of these distributions is that throughout each contract iteration, the relative bet frequencies of these gamblers remained relatively constant. Another interesting feature of

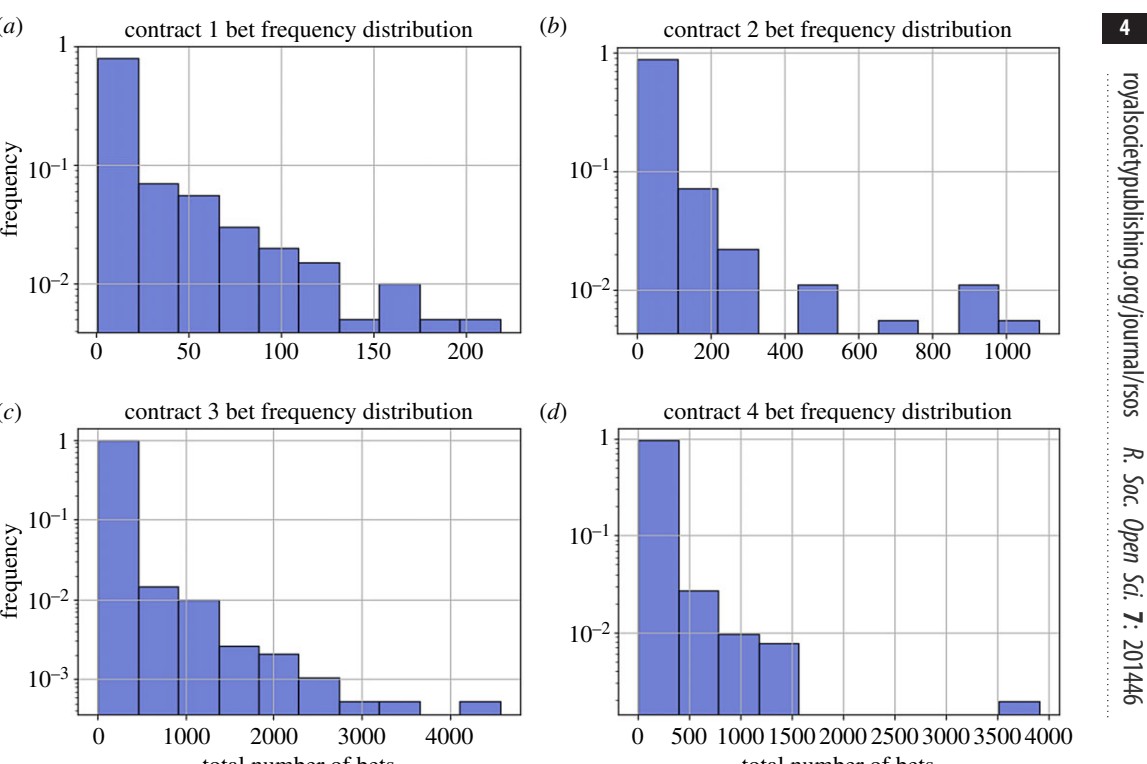

**Figure 1.** Distributions of total bet numbers for each of the four contracts, panels (a–d), from 17 April 2017 to 12 December 2017. Most gamblers have few bets while there exist some 'whale' gamblers who bet thousands of times during this time window. Table 1 for details.

**Table 1.** Total bet distribution of gamblers.

| number of total bets | contract 1 (%) | contract 2 (%) | contract 3 (%) | contract 4 (%) |
| --- | --- | --- | --- | --- |
| 1–10 | 66.16 | 62.57 | 55.16 | 60.47 |
| 10–100 | 29.85 | 24.02 | 31.87 | 26.36 |
| >100 | 3.98 | 18.99 | 12.86 | 12.98 |

the data is the existence of a tail of gamblers who bet at high frequency. The 'whale bettors', or bettors who bet more than a hundred bets and contribute most of the actual bets on the website comprise only a small fraction of the actual gamblers. Due to the gambler's ruin theory, we see that these whale bettors, who frequently gamble, must be more risk-taking. By contrast, the gamblers who gamble less must be more risk averse.

We see a very similar right skew in the bet size distributions of contracts 1, 2, 3 and 4 (figure 2). However, contract 1 displays a surprising amount of gamblers that are willing to gamble at large bet sizes (figure 2a). Additionally, there are always a few gamblers willing to bet at significant sizings (greater than 80 ETH, as shown in figure 2a–d). Possible reasons for this were probably due to the relatively low price of Ethereum (approx. 1 ETH : 50 USD). Additionally, there were only 90 000 total transactions on the Ethereum network at the time. Many of these gamblers probably did not expect the prices to exponentially rise to 500 USD/ETH.

## 2.3. Winning probability distributions

In observing the overall distribution of the probabilities that the gamblers on Etheroll gamble at, we observe two interesting fixations (figure 3). First, gamblers are extremely drawn to probabilities within the bound of $p = 0.4$–$0.6$. This is slightly different from what median cumulative prospect theory preferences specify [14], as probabilities around 0.35–0.6 are underweighted, rather than

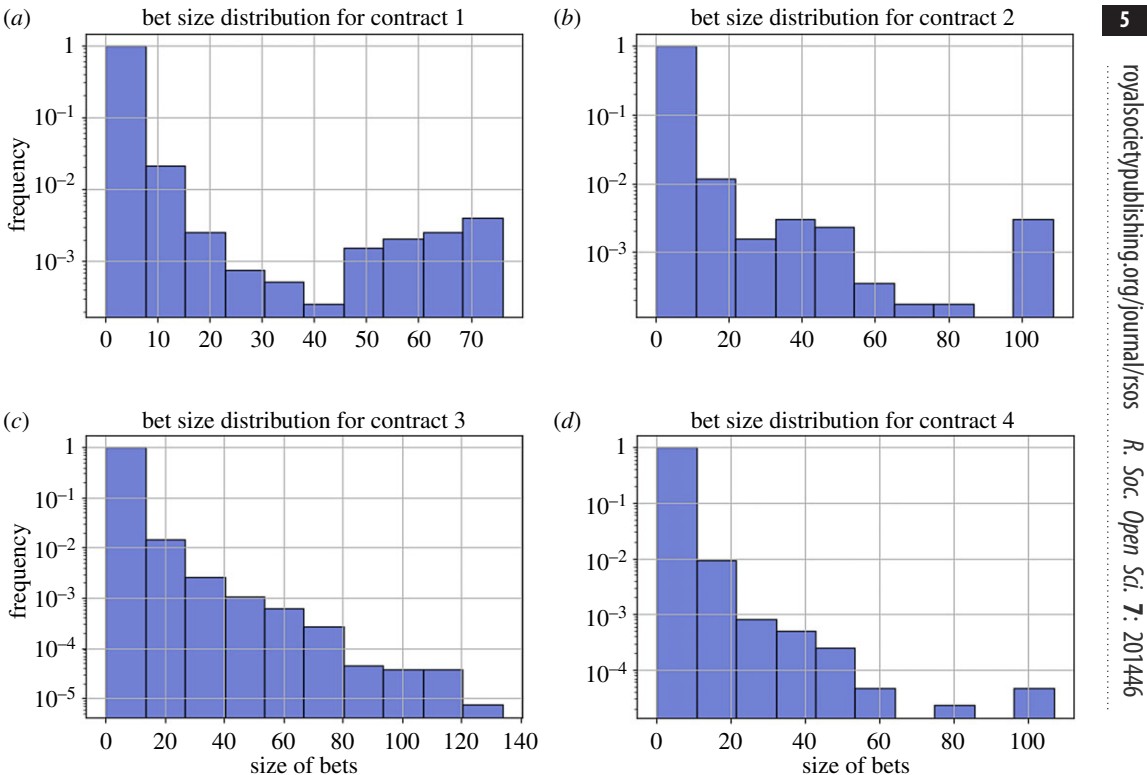

**Figure 2.** Distributions of bet sizes for each of the four contracts, panels (a–d), from 17 April 2017 to 12 December 2017. There exist 'whale bettors' who gamble with considerably large bet sizes.

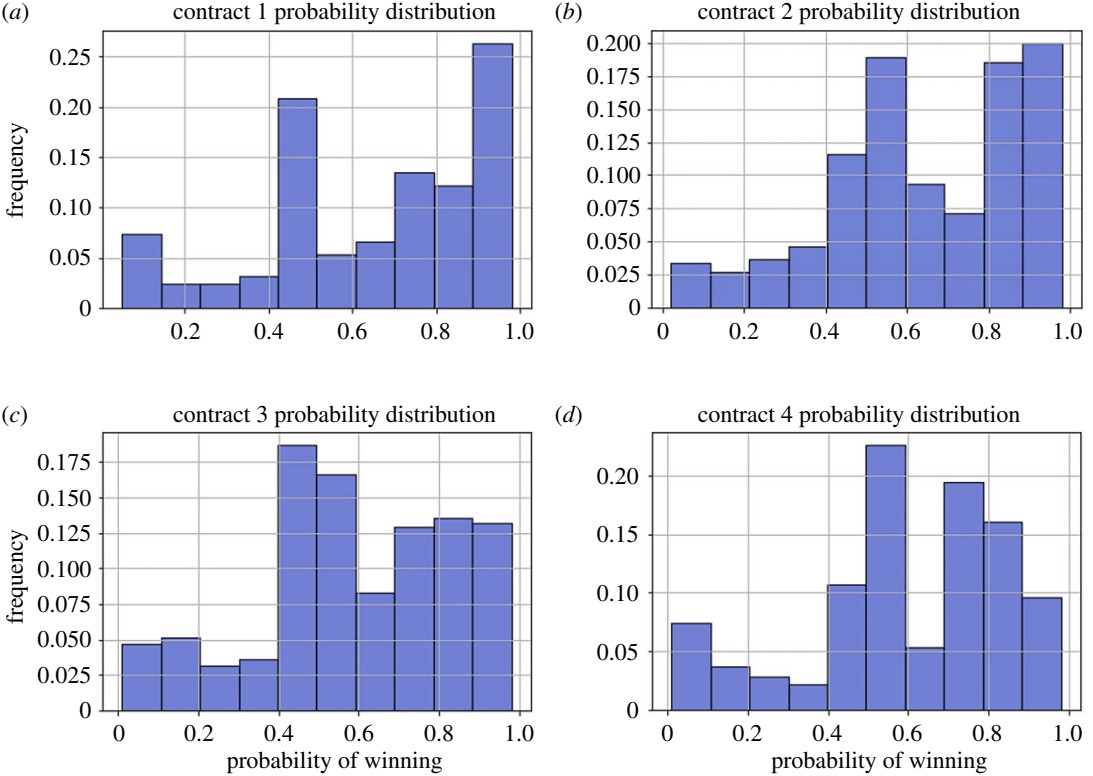

**Figure 3.** Distributions of winning probabilities of bets for each of the four contracts, panels (a–d), from 17 April 2017 to 12 December 2017. Majority of the gamblers tuned their winning probabilities within the range $p = 0.4 - 0.6$.

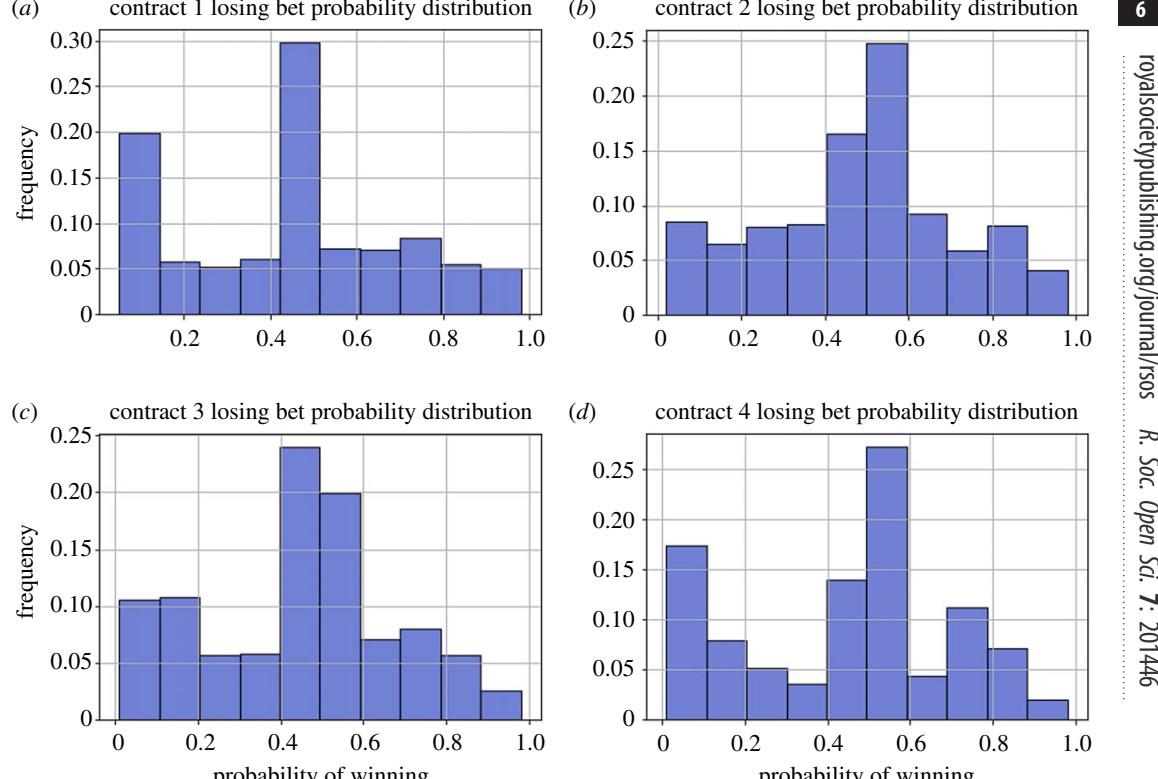

**Figure 4.** Distributions of winning probabilities of bets that ended up losing for each of the four contracts, panels (*a–d*), from 17 April 2017 to 12 December 2017. Gambling with small winning probabilities imposes great risk of losing, and thus losing bets are skewed towards unfavourable winning probabilities that are less than 50%.

overweighted. Lower probabilities have the opposite pattern. Additionally, these gamblers also have a fixation towards probabilities with very high chances of winning, within $p = 0.8$–$0.99$. This showcases these gamblers are qualitatively more risk averse. This is an odd result. First, we observe that theoretically, gamblers in this casino are more likely to be a self-selecting, risk-seeking group. Second, these gamblers must have some interest in Ethereum, and are also forced to bet significant minimum bet sizings (5–53 USD).

Additionally, we plot the probability distributions in two separate cohorts of the gambling population conditional on: gamblers who win and gamblers who lose. To do this, we segment our data into gambles of gamblers who lose and those who win.

In all four contracts, the losing cohort of gamblers have very similar losing distributions (figure 4). In general, there is a large central mean at $p = 0.5$. In contracts 2 and 3, there is a nearly normal distribution in their probabilities. We observe that in every contract, nearly 25% of bets are losing bets at around $p = 0.5$. Additionally, many of the extremely risky gamblers who bet at $p < 0.5$ are represented in this cohort. Naturally, gamblers who bet like this will tend to lose more often. The other tail end of the distribution comprises of the gamblers who take $p > 0.5$ gambles, tending to be less risky and loss averse. However, as $p \neq 1$, they are still bound to lose, and with a maximum probability of 0.99, they will still lose at least 1 out of 100 times on average.

Lastly, we observe that the winning cohort also has relatively similar general distribution throughout the four contracts (figure 5). This is due to the fact that with a large enough sample, individuality is mostly cancelled out. At first glance, it is apparent that there is a distinct left skew in the distribution, with a large amount of bets being distributed at $p > 0.5$. However, there is still a noticeable fixation by these gamblers to bet at $p = 0.5$. We also note that these winners probably tend to be more risk averse, as most of the data is accumulated at $p > 0.7$. Very few winners occur in the region of $p < 0.5$, which comprises less than 10% of the data on average per each contract iteration. Lastly, an interesting feature in almost every distribution is an aversion to $0.5 < p < 0.7$. This may be due to the shape of the perceived probability weighting function, where probabilities that are higher than $p > 0.5$ are overweighted. An explanation for why $p > 0.7$ is so popular comes in the loss-aversion formulation of

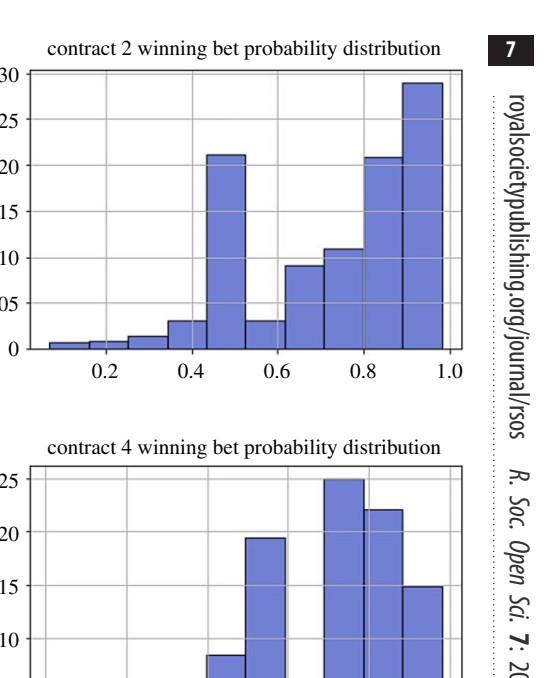

**Figure 5.** Distributions of winning probabilities of bets that ended up winning for each of the four contracts, panels (a–d), from 17 April 2017 to 12 December 2017. Gambling with favourable winning probabilities makes the distributions of winning bets heavily skewed towards these above 70%.

the value function (see more detailed analysis in our Discussion and conclusion section below). As gamblers generally wish to avoid losses, they tune their probabilities very high to avoid losses.

## 2.4. Probability versus wager sizing

Another way we can observe risk attitudes is in evaluating the relative bet sizing of gamblers versus their tuned probabilities of winning (figure 6).

These plots showcase the loss aversion of most gamblers. When staking very large bets, these gamblers exclusively bet at very high probabilities (figure 6). The largest bets are always bet at extremely high probabilities. With smaller sizings, we see a whole range of probabilities of winning. In general, this is not an extremely surprising finding, as we expect most gamblers to be loss averse.

## 2.5. Cumulative profit distributions

Another interesting function of this is that these bet frequency and probability distributions mostly shared similar shapes throughout all four contract iterations (figures 1 and 2). However, the actual value of these bets greatly varied due to the drastic changes in the market price of Ethereum, the cryptocurrency used in the gambling.

Additionally, it is interesting to see the distribution of the cumulative profits at the end of each gambler's gambling time (figure 7).

It appears that contract 1 and contract 4 have a distinctly normal distribution with $\mu < 0$. However, contract 3 seems to have a left skewed distribution, and contract 2 seems to have a right skewed distribution. We also can conclude that most gamblers do not really win anything (matching up with the fact that 60% of gamblers are recreational gamblers). We see that the mathematical edge of the casino (the house commission fee charging $e$ proportion of each bet) has effectively shifted the normal distribution to the left, as expected. This implies that most gamblers are net losers, as expected from an edged game.

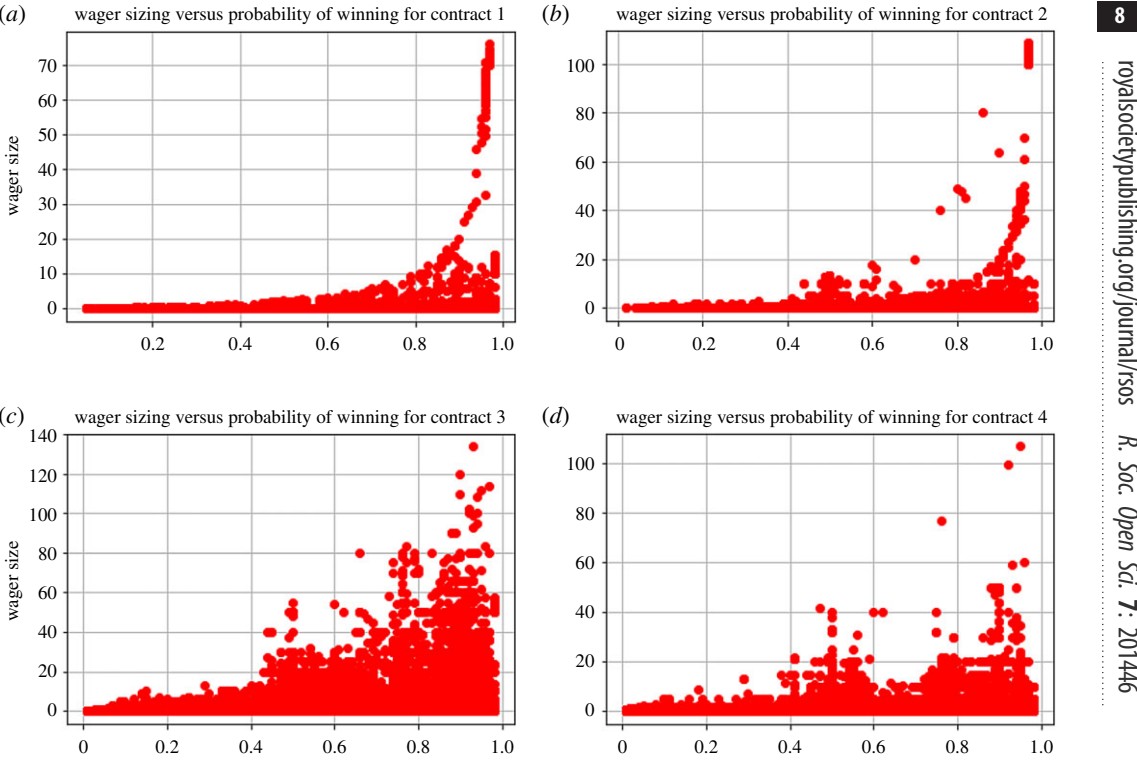

**Figure 6.** Scatter plots of wager sizing versus probability of winning for every single bet for each of the four contracts, panels (*a–d*), from 17 April 2017 to 12 December 2017. Gamblers tend to be loss averse by tuning favourable winning probabilities for large bet sizes.

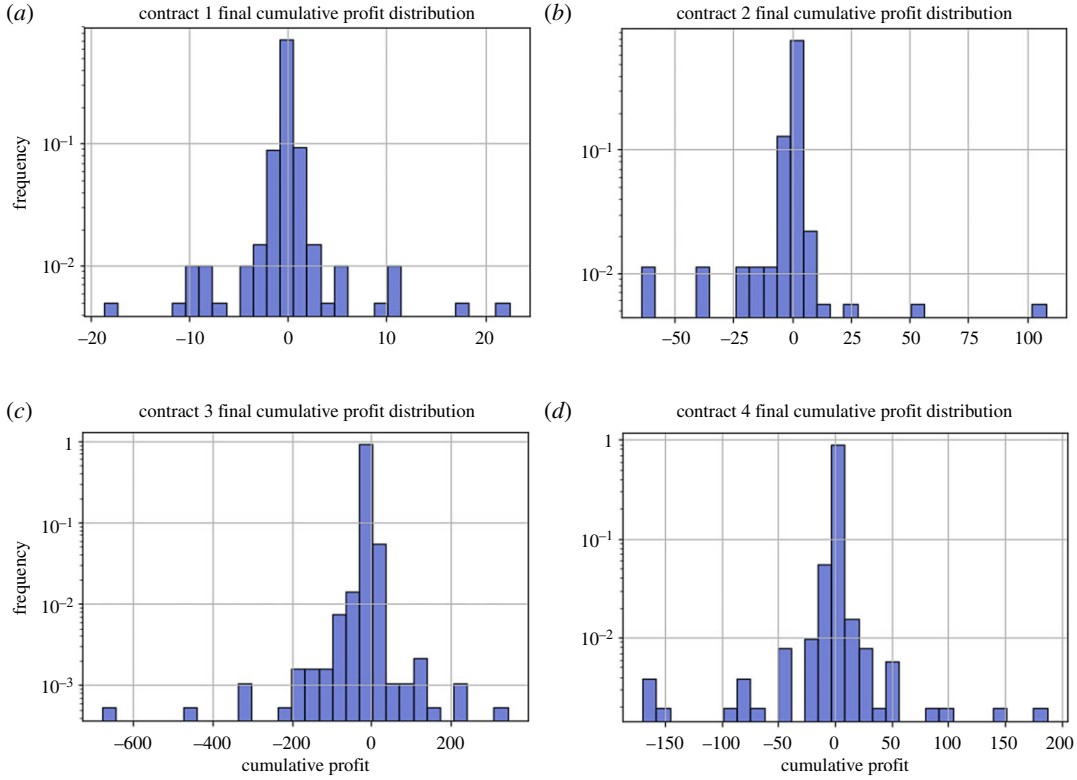

**Figure 7.** Distributions of cumulative profits for each gambler when exiting the online casino, that is, at the end of their consecutive gambling time, for each of the four contracts, panels (*a–d*), from 17 April 2017 to 12 December 2017. The mean cumulative profit is (*a*) −0.006, (*b*) −0.537, (*c*) −2.880 and (*d*) −0.658.

## 2.6. Conditioning

Another important way we can understand gambling behaviour is to look at the naturally conditional behaviour these gamblers bet with. In this regard, we shall be able to quantify the levels of risk-aversion and risk-seeking behaviours that a gambler takes, given only their previous bet. This analysis only focuses on gamblers who have betted more than once in a given contract.

Let us define $B = \{b_1, b_2, \ldots, b_n\}$, $P = \{p_1, p_2, \ldots, p_n\}$, $R = \{1, 0\}$, where $B$ is the set of ordered set of bets the gambler takes, $P$ is the ordered set of their chosen probabilities of winning and $R$ is the final result, where 0 is a loss and 1 is a win. Let us define this dice game as the mapping of all ordered tuples in $B$ and $P$ to $R$, or $B, P \to R$. Let us define

$$\forall i, i < n, b_{i+1} - b_i$$
$$\forall i, i < n, p_{i+1} - p_i.$$

This is the absolute difference in both their bet size or betting probability. We will also look at the relative (per cent) difference in the bet size

$$\forall i, i < n, \frac{b_{i+1} - b_i}{b_i}.$$

We will condition both of these on the result being either a win or a loss. Additionally, these measures obviously do not make sense for any gamblers who only bet a single time, so we will only measure these measurements for gamblers who gambled at least consecutively twice in a contract.

### 2.6.1. Probabilities

When we look at conditionally chosen probabilities of the gambler, it allows us to understand how they subjectively view a prospect given a previous result. Measuring the absolute difference in probabilities allows us a better view into the population-level, subjective perception of the probabilities of their bets. We will refer to the absolute difference between the gambler's chosen consecutive bet probabilities as their probability signal.

When looking at the summary statistics (table 2), we see a division (but not extreme division) between the previous bet result being conditioned on a win or loss. We note that as expected, gamblers who lost tend to bet at slightly larger probabilities where as gamblers that won tend to gamble at slightly smaller probabilities—taking less and more risk, respectively.

When looking at the absolute changes in probabilities chosen by the gambler, conditioned on their previous gamble, we note an interesting pattern in the distribution (figure 8). It appears that the distribution has an extremely concentrated mean with fat tails (Laplacian). Calculation of the kurtosis of this distribution (controlled for contracts) shows that it is slightly leptokurtic (for all contracts except the case of contract 4 for losers—which is slightly platykurtic). This implies a fat-tailed distribution, which implies that extreme events (deviations in probability choice size between bets) are more likely than in a normal distribution.

We also provide a normal Q-Q plot (figure 9). This Q-Q plot exhibits a very interesting kinked curve—we see that there is clear overweighting in the tails and an interesting S-shaped behaviour around the origin. This has the implication that people are more willing to have extreme changes in their chosen probabilities when measured at the bet-to-bet level compared to a normal distribution. Namely, there is greater tendency in extreme behaviours, such as drastically decreasing probabilities between bets or drastically increasing probabilities between bets. The S-shaped behaviour around the origin also shows that gamblers have slight preferences for changing their probabilities very closely around a probability signal of 0, either slightly more or slightly less. Overall, we see that people generally prefer to either make very minor changes to their chosen probabilities or extremely major ones.

### 2.6.2. Wagers

When we look at conditionally chosen wagers of the gambler, it allows us to understand how they subjectively view the bet sizing of a prospect given a previous result. Measuring the absolute difference in wager sizes allows us a better view into the user-level perception of their own prospect (table 3). We also will isolate the control for each separate contract to further understand if there are significant differences between the gamblers of each contract. We will refer to the absolute difference between wagers as the gambler's *bet signal* so as to complement the analysis of bet probability signal shown in figures 10 and 11.

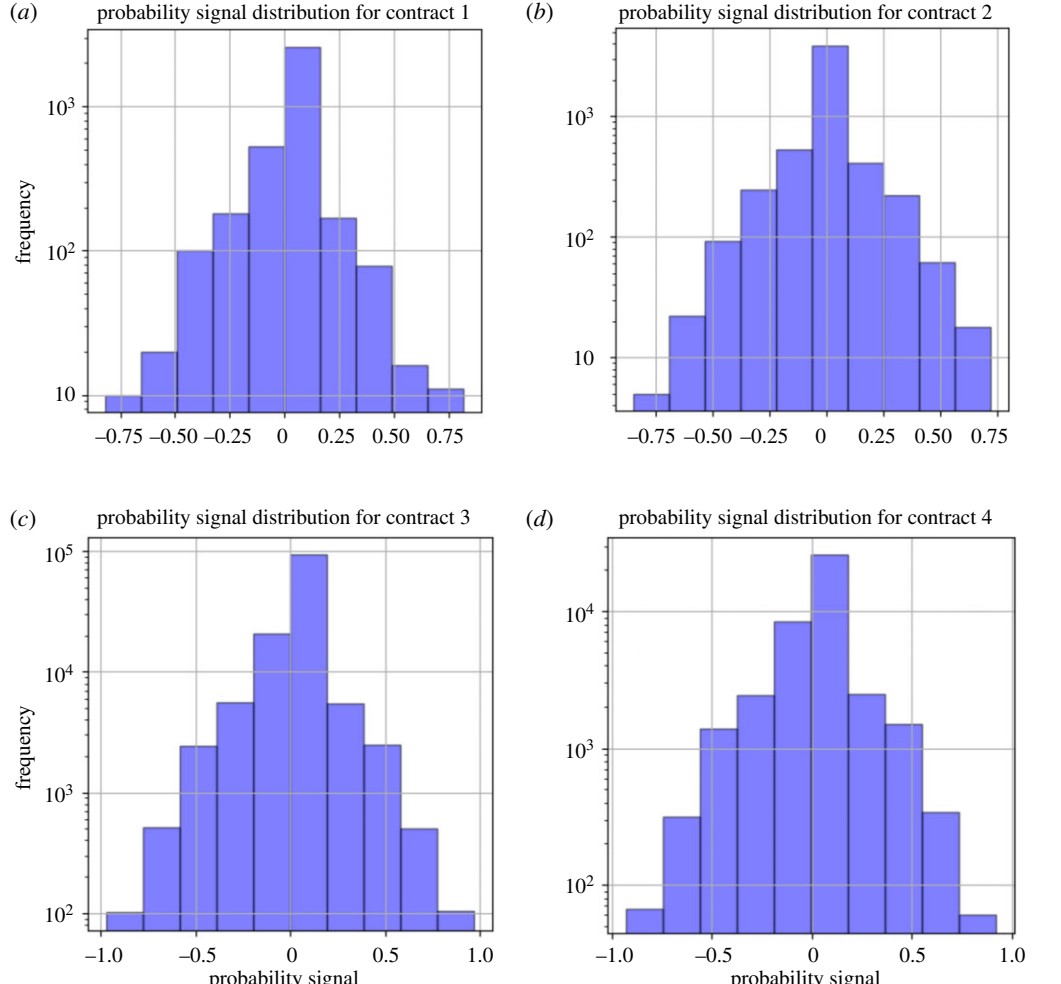

**Figure 8.** Histograms of the absolute changes in probabilities of winning tuned by gamblers conditional on previous bet outcomes (positive signs correspond to a win, and negative signs correspond to a loss) for each of the four contracts, panels (a–d), from 17 April 2017 to 12 December 2017.

**Table 2.** Summary statistics for the changes in chosen probability of winning conditional on last gambling outcomes (win or loss).

| contract type | result (previous bet) | mean | variance | skew | kurtosis |
|---|---|---|---|---|---|
| contract 1 | lost | 0.019 | 0.023 | 1.167 | 6.541 |
| | won | −0.012 | 0.021 | −0.850 | 6.261 |
| contract 2 | lost | 0.013 | 0.025 | 0.418 | 3.683 |
| | won | −0.009 | 0.019 | −0.548 | 5.259 |
| contract 3 | lost | 0.018 | 0.025 | 0.771 | 5.933 |
| | won | −0.014 | 0.019 | −0.838 | 7.681 |
| contract 4 | lost | 0.035 | 0.042 | 0.388 | 2.214 |
| | won | 0.023 | 0.025 | −0.843 | 5.281 |

When observing the summary statistics for the bet/wager signal, we note the very similar behaviour for users tending to decrease their bet sizes after a victory, but to increase their bet sizes after a loss (table 3). We also note that this distribution is heavily skewed and highly leptokurtic, no matter the cut (figure 9). An interesting difference between the contracts can be found in contract 4, which has far less variance and a generally tighter distribution.

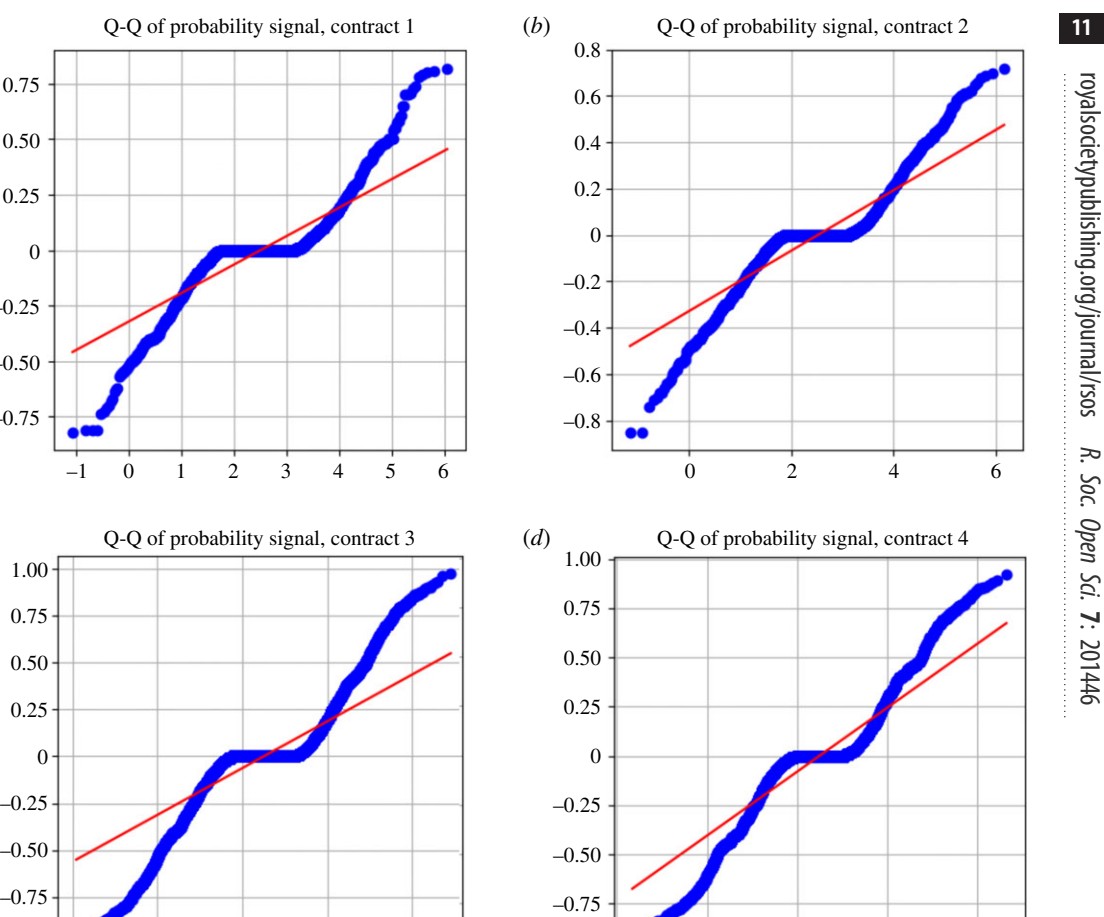

**Figure 9.** Q-Q plots of bet probability signal per contract. Shown are the Q-Q plots of the absolute changes in probabilities of winning tuned by gamblers conditional on previous bet outcomes (positive signs correspond to a win, and negative signs correspond to a loss) for each of the four contracts, panels (a–d), from 17 April 2017 to 12 December 2017. Majority of gamblers make minor changes to their chosen probabilities of winning, but it is more likely than the tail of a normal distribution that gamblers also make drastic changes at the extreme values.

When looking at the absolute changes in wagers chosen by the gambler, conditioned on their previous gamble, we note an extremely concentrated distribution. Even on the log scale, we can see an extreme concentration of values near 0. It appears that the distribution has a classic, near normal distribution. Calculation of the kurtosis of this distribution shows that it is extremely leptokurtic, implying that the existence of outliers for this distribution is far more than the normal distribution (table 3).

We also provide a Q-Q plot in figure 11. This S-shaped plot shows the classic overdispersed, leptokurtic Q-Q curve, with a large amount of data being dispersed between the left and right tails. Additionally, there is some slight S-shaped behaviour near the bet size values around 0, again implying some degree of sensitivity towards values near 0. It can be inferred that gamblers that are on the tails tend to be far more risk seeking (on the positive tail) or far more risk averse (on the negative tail), either rapidly increasing their bet sizes or rapidly decreasing their bet sizes (figure 11).

## 2.7. Gamblers of empirical interest

We are interested in looking at the existence of gambling strategies because it allows us to validate some of the ideas behind theoretical predictions. We search for strategies that are path independent and path dependent, allowing us to verify the types of gamblers in Barberis' casino model [16]. However, we will

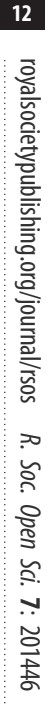

**Figure 10.** Histograms of the absolute changes in bet sizes (wagers) tuned by gamblers conditional on previous bet outcomes (positive signs correspond to a win, and negative signs correspond to a loss) for each of the four contracts, panels (*a*–*d*), from 17 April 2017 to 12 December 2017.

**Table 3.** Summary statistics for the changes in bet sizes (wager) conditional on last gambling outcomes (win or loss).

| contract type | result (previous bet) | mean | variance | skew | kurtosis |
|---|---|---|---|---|---|
| contract 1 | lost | 0.50 | 15.76 | 9.34 | 166.25 |
| | won | −0.23 | 15.07 | −11.00 | 239.89 |
| contract 2 | lost | 0.83 | 31.94 | 7.01 | 94.29 |
| | won | −0.38 | 22.67 | −9.30 | 175.02 |
| contract 3 | lost | 0.39 | 10.28 | 5.94 | 201.42 |
| | won | −0.25 | 11.25 | −6.34 | 238.12 |
| contract 4 | lost | 0.15 | 2.98 | 3.10 | 187.58 |
| | won | −0.09 | 4.23 | −3.34 | 434.36 |

observe that it is impossible for us to observe sophisticated, committed gamblers from an empirical standpoint, as we do not know what devices are used by them.

In our dataset, we are unable to evaluate gamblers that follow *proportional betting* standards, as it is not possible to parse their wallet data exactly at the times they bet. Because of this, we do not have access

**Figure 11.** Q-Q plots of the bet signal per contract. Shown are the Q-Q plots of the absolute changes in bet sizes tuned by gamblers conditional on previous bet outcomes (positive signs correspond to a win, and negative signs correspond to a loss) for each of the four contracts, panels (a–d), from 17 April 2017 to 12 December 2017. Gamblers on the tails of the distributions are more likely to chase the risk after a win (these on the right tail) and to avoid the loss after a loss (these on the left tail).

to their total wealth information, and thus we cannot search for proportions they bet at. In response to this, we will evaluate gamblers that bet at a *fixed wager* in place of proportional bets. Also of interest is the betting system in which a gambler continuously bets at a high probability (analogous to the bet everything system). This is one of the most common systems. Additionally, we will be able to evaluate gamblers who bet with negative progression, staking betting systems. We will evaluate one of the most common systems: the *martingale*. This is because gamblers following the martingale always return to their original stake, making it easier for us to detect the usage of this system. We also aim to see if gamblers have a mixture of systems, such that they transition strategies over some timescale. The reason why we choose these systems for our analysis is because it allows us to possibly observe time inconsistencies in both path-independent (fixed wager/fixed probability/high probability) and path-dependent (mixed) strategies.

To find more simple systems, such as fixed probability, fixed wager or high probability betting systems, we will use Python's pandas package for data analysis. To classify fixed systems, we convert our data, stored in csv files into pandas dataframe, and search for gamblers who have zero variance in their probability choice and wager sets (see Dataset and methods for details). To classify high probability bettors, we choose any bettor who bets only at tuning the probability of winning $p > 0.9$.

We also will only apply these methods to gamblers who have sufficient gambling data, e.g. *whale bettors* (bettors with over 100 total gambles). Additionally, we assume that the initial reference frame

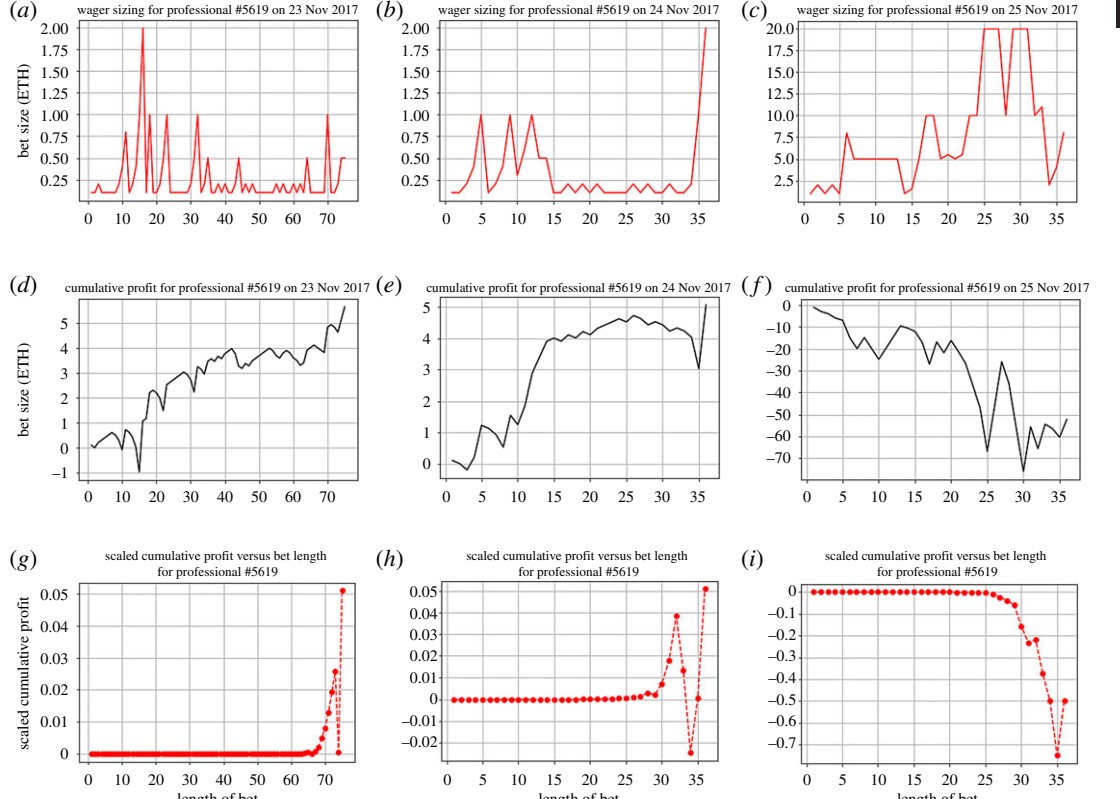

**Figure 12.** Examples of gamblers using a fixed probability strategy. Shown are (*a–c*) the wager sizes, (*d–f*) cumulative profits and (*g–i*) scaled cumulative profits over a length of consecutive gamblings, corresponding to three occasions, respectively. This gambler consistently fixed the probability of winning to be $p = 0.49$, but exhibited inconsistent exit strategies (when to stop gambling during the day).

for these gamblers, once they enter the online casino, is the beginning of the day their betting session starts at. In this way, we will be able to detect possible time inconsistencies in daily data. With this as our reference point, we found many gamblers that behave similarly to the hypothesized path-independent betting, naive gamblers who are unaware of the time inconsistencies. However, we observe some interesting path-dependent betting strategies as well (such as the martingales), or some mixture of betting strategies.

### 2.7.1. Fixed probability bettors

The most common and popular 'strategy' found in this population is the fixed probability betting strategy. In this strategy, the gambler chooses some probability, usually $p > 0.5$ and continually bets. In evaluating gamblers who bet with fixed probabilities or wagers, we can observe what happens when a gambler has no strategic time inconsistency (irrespective to exit strategies). In a very heuristic way, we can label these kinds of gamblers as more risk seeking if $p < 0.5$ and less risk seeking if $p > 0.5$.

An example of this strategy is found in the betting history of the individual labelled as professional #5619 in our dataset (figure 12). Through the three-date periods of 23 November 2017 15.17.43—23 November 2017 19.54.43, 24 November 2017 8.22.40—24 November 2017 23.30.14 and 25 November 2017 8.20.38—25 November 2017 22.04.12, this gambler gambled 147 (whale) times, all at $p = 0.49$. This is an example of a simple, path-independent strategy. No matter what his/her results are, the gambler sticks to his/her initial chosen betting strategy. Upon entering the casino, the gambler commits to this strategy, and even upon accumulating losses, this individual continues to gamble at a suboptimal probability. In gambling, at $p < 0.5$, this gambler is taking significant risk—in the long run the expectation runs negative rapidly.

This gambler may have a time inconsistency in terms of exit strategies (cf. figure 12*d–f*), but this is something we cannot empirically deduce. This individual follows the betting strategy, but exits once the losses reach some arbitrary stop-loss. We also observe loss-chasing behaviour in the third day

(figure 12*f*). This gambler begins by betting at an extremely large initial wager size of around 2 Ether (which is around the same size as the maximum bet he or she bet at in the past 2 days), and progressively increases bet sizing as losses accumulate (figure 12*c*). We observe a two-peak plateau in the bet sizing of this gambler. The first plateau is positive for the gambler. The gambler won two consecutive large bets (20 ETH), recovering a significant portion of their accumulated losses. Immediately, we observe a drastic decreasing in bet size, and a subsequent loss. The gambler continues to bet at high values, putting themselves more and more negative. After a string of consecutive losses, the gambler adjusts their bet size, eventually exiting with a loss of 41.44 Ether (figure 12*f*). Attempting to chase a few initial losses through increasing bet sizing only resulted in a larger loss. It seems that this gambler followed a gain exit pattern (figure 12*d*). Upon large wins at the end of their first sessions, the gambler stopped betting (figure 12*e*).

### 2.7.2. Fixed wager bettors

Rarer is the fixed wager strategy as seen in our dataset. This strategy is extremely simple. A gambler takes some initial stake $W_0$, and continually gambles, tuning his probability through wins and losses. Oddly, this strategy is never the only strategy the gambler employed over the timescale of a day.

### 2.7.3. Martingale bettors

As the Etheroll minimum bet size is quite large, we observe that a martingale system diverges very quickly. However, certain gamblers still follow this system. As observed, this gambler follows a martingale strategy. Their bet size starts with an initial staking size $w_0 = 0.2$ Ether, and follows

$$w_{i+1} = \begin{cases} 2^i w_0 & \text{if loss} \\ w_0 & \text{if win,} \end{cases}$$

where $w_{i+1}$ is the $(i+1)$-th bet conditioned on $i$ losses. In figure 13, we observe a very clear return to the initial staking size after every win. These graphs showcase the nearly linear staking gains of the martingale for a gambler from the timespan of 6 May 2017–7 May 2017 (where this individual betted a total of 172 times). This is because of the guaranteed return of $w_0$ from this system.

However, we note on 6 May 2017, there is a distinct drop in cumulative profits due the gambler having a stop-loss after six consecutive lost bets (figure 13*c*). This showcases the dangerous fast divergence of the martingale system. We also see an interesting time inconsistency from this gambler. Looking at the data, we see the gambler's exact deviation from this strategy at their 69th bet, where the gambler bets $2^3(0.2) = 1.6$ Ether at probability 0.5 and loses. If this gambler is following the martingale system, they should bet $2^4(0.2) = 3.2$ Ether at the same probability (to get a 1 : 1 return). However, the gambler disregards this, and gambles the same gamble of 1.6 Ether, and loses again. This causes a sharp decrease in their cumulative gains. Interestingly, the gambler makes a third bet of the same amount, and same probability. Probabilistically, this gambler will win this bet on average, but it comes with significant risk. This showcases the tendency of gamblers to chase losses—similar to the idea of gain-exit strategies. It also matches with the concept of the value function (according to the cumulative prospect theory [14], see our detailed discussion below). Theoretically, when making decisions under risk, gamblers become risk seeking when faced with losses. This helps to explain the 'loss-chasing' phenomenon.

Other possible reasons for this sudden time inconsistency in strategy could involve the gambler's total wealth (wallet size), overall bankroll. From observing their wager sizing, we observe that the gambler never bets past 1.6 Ether. Hypothetically, the next bet in the martingale sequence (3.2) could simply be too much for the gambler to continue, forcing the gambler to deviate from the planned strategy.

The trajectory of the scaled cumulative profit of this gambler shows their valuation (satisfaction) through the losses and gains, framed near the exit time (figure 13*e,f*). We observe that even though this gambler was a net positive (on 6 May 2017), their scaled cumulative profit is very faintly positive, representing the effect of the severe loss. By contrast, we see the effect of ending with a large win on 7 May 2017. The gambler's scaled cumulative profit is at a global maximum at their exit time, influencing their exit time. This gambler ended up as a net winner, winning 4.34 Ether.

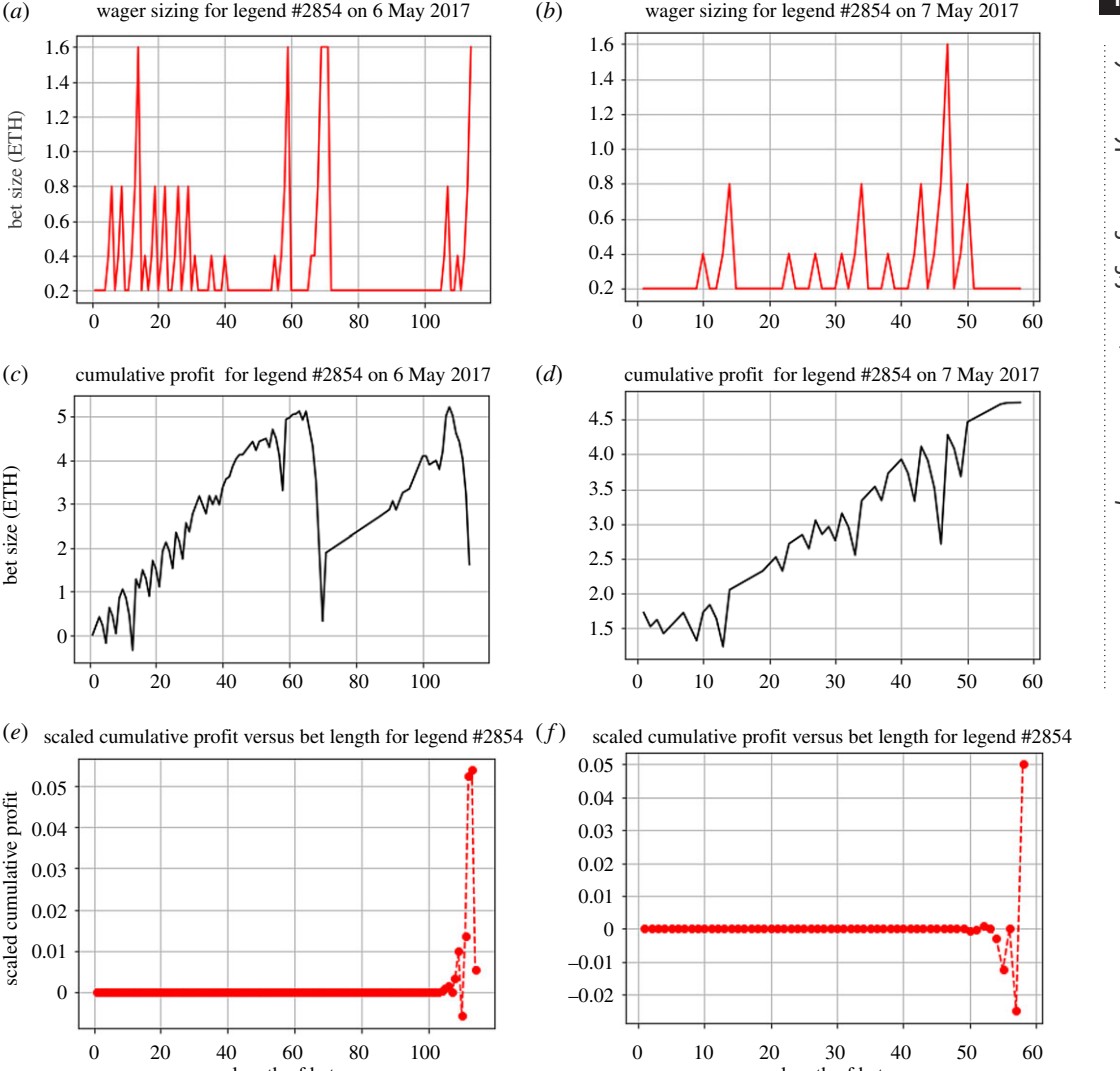

**Figure 13.** Example of martingale gamblers. Shown are (a,b) the wager sizes, (c,d) cumulative profits and (e,f) scaled cumulative profits over a varied length of consecutive gamblings, corresponding to two occasions, respectively. Martingale strategy imposes significant risk, yet can be successful.

## 3. Discussion and conclusion

One of the most well-known models for evaluating how people behave under risk situations is the cumulative prospect theory model [14]. This model is an advancement on their original prospect theory model [17], which was based on common characteristics in decision-makings, including the framing effect, nonlinear preferences, source dependence, risk-seeking behaviour and loss aversion. The 'framing effect' is the idea that humans make decisions relative to a reference point, rather than the actual result. People making decisions under risk also often exhibit a tendency towards risk-seeking behaviour, such as preferences towards low probability tail events and preferring substantial probabilities of a larger loss over sure losses (see figures 9, 11 and tables 2, 3). In tandem, loss aversion is exhibited in experiments with losses and gains. Individuals, in general, were found to be more affected by losses and gains, rather than final cumulative profit levels. Based on prior studies, Tversky and Kahneman noted a distinct asymmetry in the preferences of gamblers towards gains over losses, too significant to be attributed to risk-aversion or income effects [17]. Moreover, another important theoretical framework applicable to our work is Barberis' casino model [16], which assumes gamblers with cumulative prospect theory (CPT) preferences in the context of a casino. As Etheroll is a dice game, Barberis' casino model provides insights into understanding gamblers' choices of 'gain-

exit' versus 'loss-exit'. In [18], the authors investigate Barberis' casino model with a focus on allowing randomized and path-dependent strategies. As detailed in the Results section, we explore gambling behaviour and risk attitudes through theoretical foundations laid by these prior works.

Through our data analysis, we discover interesting patterns of gambling behaviour. One of these findings is that, as expected, a large proportion (approx. 60%) of gamblers are recreational (with 0–10 total lifetime bets). Additionally, we observe a nearly normal distribution ($\mu < 0$) of gamblers by their cumulative profit, which is expected from Etheroll's 1% house commission. Surprisingly, gamblers on Etheroll also follow a generally more loss-averse distribution of probabilities ($p > 0.5$). We also find some interesting gamblers who follow betting strategies (path-independent and path-dependent), and attempt to qualitatively explain their exit times, behaviour and risk profiles through the lens of prospect theory and cumulative profit. In looking at a mixture of strategies, we see gamblers who display strategic time inconsistencies, gain-exit and loss-exit strategies, and loss-chasing behaviour (e.g. figures 12 and 13).

To account for the individual heterogeneity in their utility function, it would be of interest to empirically approximate the cumulative prospect theory's value function [14]. The barrier of applying this method directly to the current dataset comes from the fact that many gamblers do not necessarily bet among the whole spectrum of probabilities (0, 1), but instead bet within some chosen subset. Thus, it becomes almost impossible to estimate the whole distortion function. Having an estimate of the probability distortion or value function would allow us further to quantify the risk attitudes of individuals. It would be interesting to explore other publicly available gambling datasets to see if certain gamblers have weighting functions which underweight low probabilities, but overweight high probabilities. Individuals with value functions that have steeper loss regions are more sensitive to losses, and thus more willing to take risks and chase losses. Our present study paves the way for, and will stimulate, future work in this direction.

We note that it may be promising for future work to take advantage of machine learning algorithms to search for other typical path-dependent strategies. Many other negative-progression, staking strategies exist, such as the D'Alambert, Fibonacci and Labouchere systems. However, it is very challenging to classify gamblers as gambling under these strategies, as it is quite rare for a gambler to perfectly follow a prescribed strategy. Being able to observe the deviation of gamblers from their initial strategies would pave the way to characterize the risk attitudes of the gamblers, and possible gain-exit versus loss-exit patterns. Of particular interest is to find the existence of gamblers who are clearly loss-exiting against gain-exiting. Besides, understanding individual gambling behaviour of choosing between and deploying betting systems that have varying risks from the perspective of the multi-armed bandit problem [19] is both interesting and promising.

# 4. Dataset and methods

## 4.1. Dataset

This paper entirely focuses on data collected from bets on the DApp Etheroll. Data about these bets used to be hosted on a site https://www.cryptocurrencychart.com/etheroll-live-stats. To obtain these data, we used a customized screen scraper built in Python using Requests and Beautiful Soup. Requests is a user-friendly Python library designed to handle HTTP requests, and Beautiful Soup is an HTML parsing library. We then databased these data, which consisted of approximately 250 000 individual bets and 2600 gamblers in a MySQL database. These numbers are based off the four iterations of Etheroll's smart contract. contract 1 ranges from 17 April 2017 to 24 April 2017, contract 2 ranges from 4 May 2017 to 18 May 2017, contract 3 ranges from 23 May 2017 to 25 October 2017 and contract 4 ranges from 25 October 2017 to 12 December 2017. All of these data were collected when the minimum bet on Etheroll was still 0.1 Ether (a value which ranged roughly from 4.3 USD to 52 USD).

We present simple descriptive statistics for each contract, as given in table 4.

In our dataset, a raw individual bet consists of seven fields: a Date timestamp, Player identification (such as Professional #2924, All in #2922), Bet Size (in Ether), Chance (number chosen to roll under, referred to as the probability of winning in the main text), Paid_ETH (payout in Ether) and Paid_USD (payout in USD, converted from Ether at the time). According to the maker of the website, the player identification names follow this pattern: Newbie, New address; All in, High value bets; Lucky, Wins against the odds; Play it safe, Multiple high chance bets; Against the odds, Losses with high win

**Table 4.** Environment summary statistics.

| contract | average bet size (ETH) | unique gamblers | total bets |
|---|---|---|---|
| 1 | 1.595 | 201 | 3919 |
| 2 | 2.006 | 179 | 5671 |
| 3 | 1.527 | 1889 | 132 826 |
| 4 | 0.987 | 516 | 43 425 |

chance; One in a million, Won very low chance bet; Intermediate, more than five bets; Professional, more than 25 bets; Legend, more than 100 bets.

The Chance feature is useful for determining the general riskiness of the population. Additionally, having individual player identification codes allows us to subset our data to observe the complete gambling history of each individual gambler. Having access to the timescales of the gamblers also lets us observe some interesting time inconsistencies prevalent in the data. Lastly, knowing the Bet Sizes and Payouts per bet allows us to solve for the cumulative profit of the gambler at each time step. This allows us to understand our results through the lenses of the cumulative prospect theory [14] and Barberis' casino model [16].

We subset and clean up our dataset by pre-processing the raw data. The first step is to subset the dataset in order to organize the data by individual gamblers. Next, we take these individual gamblers and subset them by the days they gambled on. Lastly, we compute the simple payout $P'$ for each gamble, which for some bet $W$, result $R \in \{$loss, win$\}$ was

$$P' = \begin{cases} -W, & R = \text{loss} \\ W(1 - p - e)/p, & R = \text{win}, \end{cases}$$

where $e$ is the house commission, 0.01, and $p$ is the probability of winning the gamble.

## 4.2. Methods

We explore this dataset using a variety of common data analysis techniques. We collect, analyse and visualize data using the Python's pandas library. As mentioned above, the data were scraped from https://www.cryptocurrencychart.com/etheroll-live-stats. An alternative way to obtain this dataset would involve setting up a computer as a Geth node (Ethereum node member), and use the JSON-RPC API to repeatedly call for contract log events (where the smart contract specifies data).

First, we aim to characterize gamblers in Etheroll, through each of their lifetime bet, bet size, betting probability and cumulative profit distributions. We will determine patterns in these data by analysing histogram distributions. This will allow us to profile the average gambler's preference for probabilities of winning, and average bet frequency. Additionally, we will analyse the relationship between bet sizing and probabilities of winning.

Next, we aim to carefully characterize gamblers who follow a few prescribed types of gambling strategies. We first preselect a few path-independent (fixing of probabilities and wager size) and path-dependent strategies (martingale). To explain certain psychological behaviour, such as loss function, we will explain the behaviour in terms of prospect theory concepts such as the weighting function and value function. Additionally, we will try to qualitatively define the risk profile of gamblers, depending on strategies and gambling patterns. We do this through observing cumulative profits over time and changes in wager sizing. To characterize the local behaviour near a gambler's exit time, we use the metric of scaled cumulative profit (figures 12g–i and 13e,f). Take some set of times $T = \{1, \ldots, \tau\} \subset \mathbb{N}$, where $\tau$ is a forced exit time, and set of cumulative profits $W = \{w_0, w_1, \ldots, w_\tau\}$, where for some $t \in T$, $w_t = \sum_{i=0}^{t} r_i$, where $r_i$ is the profit at time $i$. We scale this cumulative profit, instead taking an alternative sequence $\{2^{i-\tau} w_i\}_{i=0}^{\tau}$. This is an attempt to model the memory of a gambler, with the assumption that more recent bets are weighted more significantly than bets in the past. This follows a psychological phenomena known as the primacy and recency effect, where initial and final results are overweighted in terms of importance and memory [20].

Ethics. Ethical assessment is not required prior to conducting the research reported in this paper, as the present study does not have experiments on human subjects and animals, and does not contain any sensitive and private information.

Data accessibility. Data and relevant code for this research work are stored in GitHub: https://github.com/Orangead/Understanding-Risk-Attitudes and have been archived within the Zenodo repository (doi:10.5281/zenodo.4044242): https://doi.org/10.5281/zenodo.4044242.

Authors' contributions. J.M. and F.F. conceived the project; J.M. collected the data, performed data analyses, plotted the figures and wrote the first draft of the manuscript; F.F. analysed the data and contributed to revisions of the manuscript draft. All authors gave final approval for publication.

Competing interests. We declare we have no competing interests.

Funding. J.M. is grateful for the support by the Neukom Scholars Program. F.F. thanks the generous support by the Neukom Faculty CompX grant and the VeChain-Neukom Blockchain Research grant.

Acknowledgements. The authors thank Herbert Chang and James Detweiler for helpful discussions.

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
