## [Reviewer comments · Royal Society Open Science]

Review History

RSOS-201446.R0 (Original submission)

Review form: Reviewer 1

Is the manuscript scientifically sound in its present form?

Yes

Are the interpretations and conclusions justified by the results?

Yes

Is the language acceptable?

Yes

Do you have any ethical concerns with this paper?

No

Have you any concerns about statistical analyses in this paper?

No

Recommendation?

Accept as is

Comments to the Author(s)

This is a fresh and excellent paper in which the authors study the possible role of risk preferences in economic decision-makings with the help of real data of gambling. The applied method is sound and the authors' observations are interesting - I specially like Figs.12 & 13. I can just congratulate to this work and recommend publication as is.

There is only a tiny (technical) thing which bothers my enjoyment a bit while reading the manuscript: the authors did not pay too much attention to scale the size of labels in the figures to fit to the actual size of the main text. As a result, some of the labels became so small that I had to magnify the pdf file when watching most of the figures (and switch back when continue reading the text). Sometimes the sizes of labels within the same figures are different, hence the presentation seems a bit ugly. I think such a thought-provoking results would deserve more care on presentation in the future.

Review form: Reviewer 2

Is the manuscript scientifically sound in its present form?

Yes

Are the interpretations and conclusions justified by the results?

Yes

Is the language acceptable?

Yes

Do you have any ethical concerns with this paper?

No

Have you any concerns about statistical analyses in this paper?

No

Recommendation?

Accept with minor revision (please list in comments)

Comments to the Author(s)

This work collects available data on gambling from a DApp. The obtained dataset provides a new way to study gambling strategies and the complex dynamic of risk attitudes involved in betting decisions. Through the analysis of the data, some interesting patterns of gambling behavior are found. These results are sound and insightful, which show the role of risk preferences in human financial behavior and decision-makings beyond gambling.

As technical suggestions, I would like to ask the authors one problem as below:

The label on horizontal axis presented in fig. 3, 4, or 5 is described by bet probability. I would like to know what bet probability means, and how do you get it? I think that the detailed explanation of this term should be added in the text.

Decision letter (RSOS-201446.R0)

Dear Dr Fu

On behalf of the Editors, we are pleased to inform you that your Manuscript RSOS-201446 "Understanding Gambling Behavior and Risk Attitudes Using Cryptocurrency-based Casino Blockchain Data" has been accepted for publication in Royal Society Open Science subject to minor revision in accordance with the referees' reports. Please find the referees' comments along with any feedback from the Editors below my signature.

Please submit your revised manuscript and required files (see below) no later than 7 days from today's (ie 18-Sep-2020) date. Note: the ScholarOne system will 'lock' if submission of the revision is attempted 7 or more days after the deadline. If you do not think you will be able to meet this deadline please contact the editorial office immediately.

Kind regards,
Royal Society Open Science Editorial Office
Royal Society Open Science
opscience@royalsociety.org

on behalf of Professor Matjaz Perc (Associate Editor) and Mark Chaplain (Subject Editor)
opscience@royalsociety.org

Reviewer comments to Author:
Reviewer: 1

Comments to the Author(s)

This is a fresh and excellent paper in which the authors study the possible role of risk preferences in economic decision-makings with the help of real data of gambling. The applied method is sound and the authors' observations are interesting - I specially like Figs.12 & 13. I can just congratulate to this work and recommend publication as is.

There is only a tiny (technical) thing which bothers my enjoyment a bit while reading the manuscript: the authors did not pay too much attention to scale the size of labels in the figures to fit to the actual size of the main text. As a result, some of the labels became so small that I had to magnify the pdf file when watching most of the figures (and switch back when continue reading

the test). Sometimes the sizes of labels within the same figures are different, hence the presentation seems a bit ugly. I think such a thought-provoking results would deserve more care on presentation in the future.

Reviewer: 2

Comments to the Author(s)

This work collects available data on gambling from a DApp. The obtained dataset provides a new way to study gambling strategies and the complex dynamic of risk attitudes involved in betting decisions. Through the analysis of the data, some interesting patterns of gambling behavior are found. These results are sound and insightful, which show the role of risk preferences in human financial behavior and decision-makings beyond gambling.

As technical suggestions, I would like to ask the authors one problem as below:

The label on horizontal axis presented in fig. 3, 4, or 5 is described by bet probability. I would like to know what bet probability means, and how do you get it? I think that the detailed explanation of this term should be added in the text.

===PREPARING YOUR MANUSCRIPT===

- one version identifying all the changes that have been made (for instance, in coloured highlight, in bold text, or tracked changes);
- a 'clean' version of the new manuscript that incorporates the changes made, but does not highlight them. This version will be used for typesetting.

===PREPARING YOUR REVISION IN SCHOLARONE===

Author's Response to Decision Letter for (RSOS-201446.R0)

See Appendix A.

Decision letter (RSOS-201446.R1)

Dear Dr Fu,

It is a pleasure to accept your manuscript entitled "Understanding Gambling Behavior and Risk Attitudes Using Cryptocurrency-based Casino Blockchain Data" in its current form for publication in Royal Society Open Science.

on behalf of Professor Matjaz Perc (Associate Editor) and Mark Chaplain (Subject Editor)
openscience@royalsociety.org

Appendix A

Dear Editor:

We would like to submit our revised manuscript #RSOS-201446, entitled “Understanding Gambling Behavior and Risk Attitudes Using Cryptocurrency-based Casino Blockchain Data”, (by Jonathan Meng and Feng Fu) for publication in Royal Society Open Science.

We are very pleased with the positive feedback and valuable comments from you and the two referees. We have made all suggested changes and have attempted to clarify any points of confusion. Our detailed responses to the editor’s and referees’ comments are included below. We have also included a PDF copy of the revised manuscript with all changes highlighted.

We think the manuscript has been greatly improved in the revision process and very much hope the revised manuscript is now acceptable for publication in Royal Society Open Science.

We appreciate your kind consideration and look forward to hearing from you.

Sincerely,

Feng Fu and Jonathan Meng

Point to point response to reviewers' comments

Reviewer comments to Author:

Reviewer: 1

Comments to the Author(s)

This is a fresh and excellent paper in which the authors study the possible role of risk preferences in economic decision-makings with the help of real data of gambling. The applied method is sound and the authors' observations are interesting - I specially like Figs.12 & 13. I can just congratulate to this work and recommend publication as is.

Response: We are very grateful to the referee for this endorsement of our work and for the careful attention to detail evident in the comments. We have made all the changes exactly as the referee suggested.

There is only a tiny (technical) thing which bothers my enjoyment a bit while reading the manuscript: the authors did not pay too much attention to scale the size of labels in the figures to fit to the actual size of the main text. As a result, some of the labels became so small that I had to magnify the pdf file when watching most of the figures (and switch back when continue reading the text). Sometimes the sizes of labels within the same figures are different, hence the presentation seems a bit ugly. I think such a thought-provoking results would deserve more care on presentation in the future.

Response: Thanks for this constructive comment about the figures. We have modified all the figures with proper label/font size exactly as the referee suggested. See the revised manuscript.

Reviewer: 2

Comments to the Author(s)

This work collects available data on gambling from a DApp. The obtained dataset provides a new way to study gambling strategies and the complex dynamic of risk attitudes involved in betting decisions. Through the analysis of the data, some interesting patterns of gambling behavior are found. These results are sound and insightful, which show the role of risk preferences in human financial behavior and decision-makings beyond gambling.

Response: We appreciate that this referee's careful reading and positive feedback of our work.

As technical suggestions, I would like to ask the authors one problem as below:

The label on horizontal axis presented in fig. 3, 4, or 5 is described by bet probability. I would like to know what bet probability means, and how do you get it? I think that the detailed explanation of this term should be added in the text.

Response: Thank you for pointing this out. We apologize for this mislabeling of the x-axis that had caused confusion. We have changed the labels to "Probability of winning," to be consistent with the definition in the main text (the parameter p , the *winning probability* of bets set by a gambler) and other related figures (e.g. Fig. 6).

Article submitted to journal

Subject Areas:

Physics

Keywords:

irrationality, betting strategies, risk preferences, optimal stopping

Author for correspondence:

Jonathan Meng

e-mail: jmacb49@gmail.com

Feng Fu

e-mail: fufeng@gmail.com

Understanding Gambling Behavior and Risk Attitudes Using Cryptocurrency-based Casino Blockchain Data

Jonathan Meng¹ and Feng Fu^{1,2}

¹Department of Mathematics, Dartmouth College, Hanover, NH 03755, USA

²Department of Biomedical Data Science, Geisel School of Medicine at Dartmouth, Lebanon, NH 03756, USA

The statistical concept of Gambler's Ruin suggests that gambling has a large amount of risk. Nevertheless, gambling at casinos and gambling on the Internet are both hugely popular activities. In recent years, both prospect theory and lab-controlled experiments have been used to improve our understanding of risk attitudes associated with gambling. Despite theoretical progress, collecting real-life gambling data, which is essential to validate predictions and experimental findings, remains a challenge. To address this issue, we collect publicly available betting data from a *DApp* (decentralized application) on the Ethereum Blockchain, which instantly publishes the outcome of every single bet (consisting of each bet's timestamp, wager, probability of winning, userID, and profit). This online casino is a simple dice game that allows gamblers to tune their own winning probabilities. Thus the dataset is well suited for studying gambling strategies and the complex dynamic of risk attitudes involved in betting decisions. We analyze the dataset through the lens of current probability-theoretic models and discover empirical examples of gambling systems. Our results shed light on understanding the role of risk preferences in human financial behavior and decision-makings beyond gambling.

1. Introduction

The client server model, the most widely used computing network model in the world today, allows devices (clients) to request services or resources from other devices (servers). The client initiates a request to the server and receives a response, which usually gives the client the service or resource it requested. Some examples of this are the World Wide Web, or email. A major issue with this model is that if the server stops working, everything else also ceases functioning. Additionally, if hackers manage to break into the server, they could steal any client information (e.g., Social Security Numbers, Credit Card information) stored inside. This model inherently leads to centralization of computing power towards larger entities, such as government or multinational corporations [1].

In contrast, a peer-to-peer network lets any of its members (nodes) share information or services on the network. All nodes have equal privilege, which means any node in the network can give another node in the network a desired resource or service. The most famous example of peer-to-peer networking is in torrenting, where an initial server, called a seed, uploads a file. Nodes of the torrent network (the swarm) divide up this file into pieces and request missing pieces from other computers in the network. Once pieces are obtained by a client, or downloading node, the pieces are constructed into the original file. In this way, computing power is not monopolized; it is shared [2]. This model is both fault-tolerant (i.e., continues to work even if a single or multiple members fails), and decentralized.

Ethereum is a distributed, peer-to-peer computing network, released in 2015, that allows its nodes to conduct transactions and build applications. On the Ethereum network, the main currency, Ether, powers all peer-to-peer transactions for goods and services [3].

One of the most important features of Ethereum is its usage of blockchain technology. The blockchain is a decentralized, publicly available chain of transactions. Anyone can download software (Geth, Parity) and turn their computer into a node, or a member of the Ethereum network. The peer-to-peer nature of the network allows computing power to be evenly distributed and accessible. Because all nodes contain a copy of the blockchain, each node has access to the same information. All nodes retain perfect information and verify transactions. Through the usage of blockchain technology, Ethereum aims to shift the current paradigm of computing from the client-server model to a decentralized, peer-to-peer model.

All nodes verify transactions in order to ensure that new transactions are not fraudulent. Once enough transactions are verified, these transactions are packaged together into a block. Certain nodes, called miners, then compete to compute a difficult cryptographic hashing problem, called ETHhash. This system, which rewards miners for work done is referred to as a Proof of Work System. Once a miner solves the problem, the mined block is then added to the blockchain.

After successfully packaging a block, miners are awarded with currency that is used to pay for transactions, such as Ether, or Bitcoin. On the Ethereum network, this reward is up to 5 Ether. Because each transaction is verified by all the nodes in the network, blockchains are extremely resistant to attempts of fraudulent modification. If an attacker attempts to change the system, they would have to generate an alternate chain from scratch. According to the original white paper (specification) of Bitcoin, the block synchronization of these two parties is modeled as a binomial random walk. From this, we see that the effective probability of an attacker succeeding in creating a fraudulent blockchain approaches 0 if the attacker is more than 25 blocks behind the actual blockchain [4].

Another important feature of blockchain technology is that it allows user to user transactions to be pseudoanonymous. This is due to a hashing of the transaction IDs and their corresponding wallet IDs. This is extremely important, as it allows for transparency of data [5]. Users do not have to worry about exposing their identity to the public. In recent years, publicly available blockchain data has attracted growing interest from diverse fields to explore human behavior and in particular online transactions in a wide range of blockchain-based applications [6–10].

Ethereum has also introduced the idea of programming blockchain operations through a technology called the smart contract. A smart contract is an automated script written in Ethereum's own scripting language, Solidity, that allows an individual to exchange a specified good or service. A popular comparison for smart contracts is the vending machine. If a user of the smart contract gives the vending machine a certain fee, and a product comes out. Accordingly, if a user inputs some cryptocurrency into a smart contract, it executes an exchange of goods or services. As smart contracts are also automated, they erase the need for a middleman. Smart contracts, if programmed properly, can be used for a variety of applications, such as vote automation or tax collection. Building an application on top of a smart contract creates a decentralized application (DApp). A DApp is completely decentralized (no single owner) and automated by its associated smart contract. Currently, there are around 1,539 DApps on the Ethereum Blockchain [11].

We study the behavioral dynamics of gamblers on a DApp known as Etheroll. Etheroll simulates a virtual dice gambling game where all bets are made in Ether and published on the Ethereum blockchain. Etheroll has an associated smart contract on the Ethereum network which specifies house edges, payouts, and dividends to investors [12]. To begin the dice game, the gambler chooses a number between 2 and 99 (inclusive). The probability that the gambler wins is the number he or she chooses, minus 1, meaning that the gambler can choose between a 1% to 98% chance of winning. The payout (P') formula, if the house commission per bet is $e = 1\%$, probability of winning is p , and initial wager is W is:

$$P' = W \left(\frac{1 - p - e}{p} \right).$$

The smart contract then simulates a hundred-sided dice roll. If the result of the dice roll is any number smaller than the number the gambler chose, the gambler wins. After the transaction between the smart contract and the gambler processes, the gambler receives a payout (in Ether) directly to their Ethereum wallet which is inversely proportional to the probability they bet at. Naturally, lower probabilities of winning have higher payouts, and higher probabilities of winning have lower payouts. Regardless of their chosen winning probabilities p , the expected payout of gamblers is negative, $E[P'] = -eW$, due to the house commission fee charging e percentage for each bet W .

These transactions are publicly available on the Ethereum blockchain. Due to the massive amount of verifying nodes on the Ethereum network, we can be sure about the validity of these transactions. We will explore this data for all four of Etheroll's smart contract updates from April 17th, 2017 to December 12th, 2017. Obtaining real life gambling data, especially data from gambles in casinos is very difficult, if not impossible to obtain. Because of this, mathematical models pertaining to gambling are almost entirely theoretically based. Every bet from Etheroll consists of the bet's Timestamp, Wager, Probability of Winning, UserID, and Profit. With this data, we shall empirically explore gambling behavior and risk attitudes in light of the cumulative prospect theory [13].

This dataset has many other interesting properties. Having access to timestamps allows us to identify possible changes in strategy influenced by gambling results over time, in their gambling patterns. The fact that gamblers are able to tune their own betting probabilities is also crucial. The ability to tune the effective odds in a wager allows us to evaluate probable risk profiles of certain gamblers. Additionally, we focus on characterizing the entire risk attitudes of the entire gambling ecosystem as a whole. We are also able to evaluate the existence and usage of staking gambling systems (path-dependent strategies). The unique completeness and continuity of this data also allows us to empirically evaluate some famous psychological frameworks, such as the cumulative prospect theory [13]. We also characterize and quantify the effect of a gambler's cumulative "signal", or scaled cumulative profit on their winning probability distributions and betting strategies. This scaling allows us to model the lessened effect of losses and gains over time.

2. Results

(a) Overview

This population of gamblers on the Ethereum blockchain allows us to empirically observe the tendencies of gamblers in a casino-like environment for the first time. The minimum bet-sizing of 0.1 Ether (4 - 53 USD in this dataset) simulates casino-like stakes [14]. The game these gamblers play is simple, parameterized only by the probability of winning they chose and their wager size. We will first characterize the types of gamblers in this online casino through these two variables, the overall distribution of these two variables, and the paired cohort of winning gamblers and losing gamblers. We will also look at how gamblers behave when conditioned on the previous bet. To do this, we will measure the absolute and relative changes in both the probabilities they chose and their wager sizing. We will also measure the cumulative profit of gamblers - to understand how many gamblers actually end up winning anything. Lastly, we will look at gambling strategies and gamblers of interest.

(b) Wager Sizing

To first characterize this population of gamblers, we visualize the total bet frequency distributions of each gambler. Using histograms, we track each gambler's total gambles per smart contract, and the corresponding frequency of occurrence. In doing so, there is a very pronounced right skew in the distribution of the amount of gambles of each gambler. In fact, in each smart contract iteration, the gamblers who gamble only 1 to 10 times comprise approximately 60-65% of the entire population. This heavy right skew shows that most gamblers in this DApp are mainly recreational gamblers who place anywhere from 1 to 10 bets (See Tab. 1, Fig. 1).

Table 1: Total Bet Distribution of Gamblers

Number of Total Bets	Contract 1(%)	Contract 2(%)	Contract 3(%)	Contract 4(%)
1 – 10	66.16	62.57	55.16	60.47
10 – 100	29.85	24.02	31.87	26.36
> 100	3.98	18.99	12.86	12.98

An interesting qualitative feature of these distributions is that throughout each contract iteration, the relative bet frequencies of these gamblers remained relatively constant. Another interesting feature of the data is the existence of a tail of gamblers who bet at high frequency. The “Whale Bettors”, or bettors who bet more than a hundred bets and contribute most of the actual bets on the website comprise only a small fraction of the actual gamblers. Due to the gambler's ruin theory, we see that these whale bettors, who frequently gamble, must be more risk-taking. In contrast, the gamblers who gamble less must be more risk-averse.

We see a very similar right skew in the bet size distributions of Contracts 1, 2, 3 and 4 (Fig. 2). However, Contract 1 displays a surprising amount of gamblers that are willing to gamble at large bet sizes (Fig. 2a). Additionally, there are always a few gamblers willing to bet at significant sizings (> 80 ETH, as shown in Figs. 2a-d). Possible reasons for this were probably due to the relatively low price of Ethereum (approximately 1 ETH : 50 USD). Additionally, there were only 90,000 total transactions on the Ethereum network at the time. Many of these gamblers probably did not expect the prices to exponentially rise to 500 USD/ETH.

(c) Winning Probability Distributions

In observing the overall distribution of the probabilities that the gamblers on Etheroll gamble at, we observe two interesting fixations (Fig. 3). First, gamblers are extremely drawn to probabilities

Figure 1: Distributions of total bet numbers for each of the four contracts, panels (a) - (d), from April 17th, 2017 to December 12th, 2017. Most gamblers have few bets while there exist some “whale” gamblers who bet thousands of times during this time window. Tab. 1 for details.

within the bound of $p = 0.4 - 0.6$. This is slightly different from what median cumulative prospect theory preferences specify [13], as probabilities around $0.35 - 0.6$ are underweighted, rather than overweighted. Lower probabilities have the opposite pattern. Additionally, these gamblers also have a fixation towards probabilities with very high chances of winning, within $p = 0.8 - 0.99$. This showcases these gamblers are qualitatively more risk averse. This is an odd result. First, we observe that theoretically, gamblers in this casino are more likely to be a self-selecting, risk seeking group. Second, these gamblers must have some interest in Ethereum, and are also forced to bet significant minimum bet sizings (5-53 USD).

Additionally, we plot the probability distributions in two separate cohorts of the gambling population conditional on: gamblers who win, and gamblers who lose. To do this, we segment our data into gambles of gamblers who lose and those who win.

In all four contracts, the losing cohort of gamblers have very similar losing distributions (see Fig. 4). In general, there is a large central mean at $p = 0.5$. In Contracts 2 and 3, there is a nearly normal distribution in their probabilities. We observe that in every contract, nearly 25% of bets are losing bets at around $p = 0.5$. Additionally, many of the extremely risky gamblers who bet at $p < 0.5$ are represented in this cohort. Naturally, gamblers who bet like this will tend to lose more often. The other tail end of the distribution comprises of the gamblers take $p > 0.5$ gamblers, tending to be less risky and loss averse. However, as $p \neq 1$, they are still bound to lose, and with a maximum probability of 0.99, they will still lose at least 1 out of 100 times on average.

Lastly, we observe that the winning cohort also has relatively similar general distribution throughout the four contracts (Fig. 5). This is due to the fact that with a large enough sample, individuality is mostly canceled out. At first glance, it is apparent that there is a distinct left skew in the distribution, with a large amount of bets being distributed at $p > 0.5$. However, there is still a noticeable fixation by these gamblers to bet at $p = 0.5$. We also notice that these winners

Figure 2: Distributions of bet sizes for each of the four contracts, panels (a) - (d), from April 17th, 2017 to December 12th, 2017. There exist “whale bettors” who gamble with considerably large bet sizes.

probably tend to be more risk averse, as most of the data is accumulated at $p > 0.7$. Very few winners occur in the region of $p < 0.5$, which comprises less than 10% of the data on average per each contract iteration. Lastly, an interesting feature in almost every distribution is an aversion to $0.5 < p < 0.7$. This may be due to the shape of the perceived probability weighting function, where probabilities that are higher than $p > 0.5$ are overweighted. An explanation for why $p > 0.7$ is so popular comes in the loss-aversion formulation of the value function (see more detailed analysis in our Discussions section below). As gamblers generally wish to avoid losses, they tune their probabilities very high to avoid losses.

(d) Probability vs Wager Sizing

Another way we can observe risk attitudes is in evaluating the relative bet sizing of gamblers versus their tuned probabilities of winning (Fig. 6).

These plots showcase the loss aversion of most gamblers. When staking very large bets, these gamblers exclusively bet at very high probabilities (Fig. 6). The largest bets are always bet at extremely high probabilities. With smaller sizings, we see a whole range of probabilities of winning. In general, this is not an extremely surprising finding, as we expect most gamblers to be loss averse.

(e) Cumulative Profit Distributions

Another interesting function of this is that these bet frequency and probability distributions mostly shared similar shapes throughout all four contract iterations (Fig. 1, and Fig. 2). However,

Figure 3: Distributions of winning probabilities of bets for each of the four contracts, panels (a) - (d), from April 17th, 2017 to December 12th, 2017. Majority of the gamblers tuned their winning probabilities within the range $p = 0.4 - 0.6$.

the actual value of these bets greatly varied due to the drastic changes in the market price of Ethereum, the cryptocurrency used in the gambling.

Additionally, it is interesting to see the distribution of the cumulative profits at the end of each gambler's gambling time (see Fig. 7).

It appears that Contract 1, Contract 4 have a distinctly normal distribution with $\mu < 0$. However, Contract 3 seems to have a left skewed distribution, and Contract 4 seems to have a right skewed distribution. We also can conclude that most gamblers do not really win anything (matching up with the fact that 60% of gamblers are recreational gamblers). We see that the mathematical edge of the casino (the house commission fee charging e proportion of each bet) has effectively shifted the normal distribution to the left, as expected. This implies that most gamblers are net losers, as expected from an edged game.

(f) Conditioning

Another important way we can understand gambling behavior is to look at the naturally conditional behavior these gamblers bet with. In this regard, we shall be able to quantify the levels of risk aversion and risk seeking behaviors that a gambler takes, given only their previous bet. This analysis only focuses on gamblers who have betted more than once in a given contract.

Let us define $B = \{b_1, b_2, \dots, b_n\}$, $P = \{p_1, p_2, \dots, p_n\}$, $R = \{1, 0\}$ where B is the set of ordered set of bets the gambler takes, P is the ordered set of their chosen probabilities of winning, and R is the final result, where 0 is a loss and 1 is a win. Let us define this dice game as the mapping of all ordered tuples in B and P to R , or $B, P \rightarrow R$. Let us define:

- $\forall i, i < n, b_{i+1} - b_i$

Figure 4: Distributions of winning probabilities of bets that ended up losing for each of the four contracts, panels (a) - (d), from April 17th, 2017 to December 12th, 2017. Gambling with small winning probabilities imposes great risk of losing, and thus losing bets are skewed towards unfavorable winning probabilities that are less than 50%.

- $\forall i, i < n, p_{i+1} - p_i$

This is the absolute difference in both their bet size or betting probability. We will also look at the relative (percent) difference in the bet size:

- $\forall i, i < n, \frac{b_{i+1} - b_i}{b_i}$

We will condition both of these on the result being either a win or a loss. Additionally, these measures obviously do not make sense for any gamblers who only bet a single time, so we will only measure these measurements for gamblers who gambled at least consecutively twice in a contract.

(i) Probabilities

When we look at conditionally chosen probabilities of the gambler, it allows us to understand how they subjectively view a prospect given a previous result. Measuring the absolute difference in probabilities allows us a better view into the population level, subjective perception of the probabilities of their bets. We will refer to the absolute difference between the gambler's chosen consecutive bet probabilities as their probability signal.

When looking at the summary statistics (Tab. 2), we see a division (but not extreme division) between the previous bet result being conditioned on a win or loss. We notice that as expected, gamblers who lost tend to bet at slightly larger probabilities where as gamblers that won tend to gamble at slightly smaller probabilities – taking less and more risk, respectively.

Figure 5: Distributions of winning probabilities of bets that ended up winning for each of the four contracts, panels (a) - (d), from April 17th, 2017 to December 12th, 2017. Gambling with favorable winning probabilities makes the distributions of winning bets heavily skewed towards these above 70%.

Table 2: Summary statistics for the changes in chosen probability of winning conditional on last gambling outcomes (win or loss).

Contract Type	Result (Previous Bet)	Mean	Variance	Skew	Kurtosis
Contract 1	lost	0.019	0.023	1.167	6.541
	won	-0.012	0.021	-0.850	6.261
Contract 2	lost	0.013	0.025	0.418	3.683
	won	-0.009	0.019	-0.548	5.259
Contract 3	lost	0.018	0.025	0.771	5.933
	won	-0.014	0.019	-0.838	7.681
Contract 4	lost	0.035	0.042	0.388	2.214
	won	0.023	0.025	-0.843	5.281

When looking at the absolute changes in probabilities chosen by the gambler, conditioned on their previous gamble, we notice an interesting pattern in the distribution (Fig. 10). It appears that the distribution has an extremely concentrated mean with fat tails (Laplacian). Calculation of the kurtosis of this distribution (controlled for contracts) shows that it is slightly leptokurtic (for all contracts except the case of contract 4 for losers - which is slightly platykurtic). This implies a fat tailed distribution, which implies that extreme events (deviations in probability choice size between bets) are more likely than in a normal distribution.

Figure 6: Scatter plots of wager sizing versus probability of winning for every single bet for each of the four contracts, panels (a) - (d), from April 17th, 2017 to December 12th, 2017. Gamblers tend to be loss averse by tuning favorable winning probabilities for large bet sizes.

We also provide a normal Q-Q plot (Fig. i). This Q-Q plot exhibits a very interesting kinked curve – we see that there is clear overweighting in the tails and an interesting S-shaped behavior around the origin. This has the implication that people are more willing to have extreme changes in their chosen probabilities when measured at the bet to bet level compared to a normal distribution. Namely, there is greater tendency in extreme behaviors, such as drastically decreasing probabilities between bets or drastically increasing probabilities between bets. The S-shaped behavior around the origin also shows that gamblers have slight preferences for changing their probabilities very closely around a probability signal of 0, either slightly more or slightly less. Overall, we see that people generally prefer to either make very minor changes to their chosen probabilities or extremely major ones.

(ii) Wagers

When we look at conditionally chosen wagers of the gambler, it allows us to understand how they subjectively view the bet sizing of a prospect given a previous result. Measuring the absolute difference in wager sizes allows us a better view into the user level perception of their own prospect (Tab. 3). We also will isolate the control for each separate contract to further understand if there is significant differences between the gamblers of each contract. We will refer to the absolute difference between wagers as the gambler's *bet signal* so as to complement the analysis of bet probability signal shown in Figs. 10 and i.

When observing the summary statistics for the bet/wager signal, we notice the very similar behavior for users tending to decrease their bet sizes after a victory, but to increase their bet sizes after a loss (Tab. 3). We also notice that this distribution is heavily skewed and highly leptokurtic,

Figure 7: Distributions of cumulative profits for each gambler when exiting the online casino, that is, at the end of their consecutive gambling time, for each of the four contracts, panels (a) - (d), from April 17th, 2017 to December 12th, 2017. The mean cumulative profit is (a) -0.006 , (b) -0.537 , (c) -2.880 , (d) -0.658 .

Table 3: Summary statistics for the changes in bet sizes (wager) conditional on last gambling outcomes (win or loss).

Contract Type	Result (Previous Bet)	Mean	Variance	Skew	Kurtosis
Contract 1	lost	0.50	15.76	9.34	166.25
	won	-0.23	15.07	-11.00	239.89
Contract 2	lost	0.83	31.94	7.01	94.29
	won	-0.38	22.67	-9.30	175.02
Contract 3	lost	0.39	10.28	5.94	201.42
	won	-0.25	11.25	-6.34	238.12
Contract 4	lost	0.15	2.98	3.10	187.58
	won	-0.09	4.23	-3.34	434.36

no matter the cut (Fig. 10). An interesting difference between the contracts can be found in contract 4, which has far less variance and a generally tighter distribution.

When looking at the absolute changes in wagers chosen by the gambler, conditioned on their previous gamble, we notice an extremely concentrated distribution. Even on the log scale, we can see an extreme concentration of values near 0. It appears that the distribution has a classic, near normal distribution. Calculation of the kurtosis of this distribution shows that it is extremely leptokurtic, implying that the existence of outliers for this distribution is far more than the normal distribution (Tab. 3).

Figure 8: Histograms of the absolute changes in probabilities of winning tuned by gamblers conditional on previous bet outcomes (positive signs correspond to a win, and negative signs correspond to a loss) for each of the four contracts, panels (a) - (d), from April 17th, 2017 to December 12th, 2017.

We also provide a Q-Q plot in Fig. 11. This S-shaped plot shows the classic overdispersed, leptokurtic Q-Q curve, with a large amount of data being dispersed between the left and right tails. Additionally, there is some slight S-shaped behavior near the bet size values around 0, again implying some degree of sensitivity towards values near 0. It can be inferred that gamblers that are on the tails tend to be far more risk seeking (on the positive tail) or far more risk adverse (on the negative tail), either rapidly increasing their bet sizes or rapidly decreasing their bet sizes (Fig. 11).

Figure 9: Q-Q plots of bet probability signal per contract. Shown are the Q-Q plots of the absolute changes in probabilities of winning tuned by gamblers conditional on previous bet outcomes (positive signs correspond to a win, and negative signs correspond to a loss) for each of the four contracts, panels (a) - (d), from April 17th, 2017 to December 12th, 2017. Majority of gamblers make minor changes to their chosen probabilities of winning, but it is more likely than the tail of a normal distribution that gamblers also make drastic changes at the extreme values.

(g) Gamblers of Empirical Interest

We are interested in looking at the existence of gambling strategies because it allows us to validate some of the ideas behind theoretical predictions. We search for strategies that are path independent and path dependent, allowing us to verify the types of gamblers in the Barberis' Casino model [15]. However, we will observe that it is impossible for us to observe sophisticated, committed gamblers from an empirical standpoint, as we do not know what devices used by them.

In our dataset, we are unable to evaluate gamblers that follow *proportional betting* standards, as it is not possible to parse their wallet data exactly at the times they bet at. Because of this, we do not have access to their total wealth information, and thus we cannot search for proportions they bet at. In response to this, we will evaluate gamblers that bet at a *fixed wager* in place of proportional bets. Also of interest is the betting system in which a gambler continuously bets at a high probability (analogous to the bet everything system). This is one of the most common

Figure 10: Histograms of the absolute changes in bet sizes (wagers) tuned by gamblers conditional on previous bet outcomes (positive signs correspond to a win, and negative signs correspond to a loss) for each of the four contracts, panels (a) - (d), from April 17th, 2017 to December 12th, 2017.

systems. Additionally, we will be able to evaluate gamblers who bet with negative progression, staking betting systems. We will evaluate one of the most common systems: the *martingale*. This is because gamblers following the martingale always return to their original stake, making it easier for us to detect the usage of this system. We also aim to see if gamblers have a mixture of systems, such that they transition strategies over some timescale. The reason why we choose these systems for our analysis is because it allows us to possibly observe time inconsistencies in both path-independent (fixed wager/fixed probability/high probability) and path-dependent (mixed) strategies.

To find more simple systems, such as fixed probability, fixed wager or high probability betting systems, we will use Python’s Pandas package for data analysis. To classify fixed systems, we convert our data, stored in csv files into Pandas Dataframe, and search for gamblers who have zero variance in their probability choice and wager sets (see Dataset and Methods for details). To classify high probability bettors, we choose any bettor who bets only at tuning the probability of winning $p > 0.9$.

Figure 11: Q-Q plots of the bet signal per contract. Shown are the Q-Q plots of the absolute changes in bet sizes tuned by gamblers conditional on previous bet outcomes (positive signs correspond to a win, and negative signs correspond to a loss) for each of the four contracts, panels (a) - (d), from April 17th, 2017 to December 12th, 2017. Gamblers on the tails of the distributions are more likely to chase the risk after a win (these on the right tail) and to avoid the loss after a loss (these on the left tail).

We also will only apply these methods to gamblers who have sufficient gambling data, e.g., *whale bettors* (bettors with over 100 total gambles). Additionally, we assume that the initial reference frame that these gamblers once they enter the online casino is the beginning of the day their betting session starts at. In this way, we will be able to detect possible time inconsistencies in daily data. With this as our reference point, we found many gamblers that behave similarly to the hypothesized path-independent betting, naive gamblers who are unaware of the time inconsistencies. However, we observe some interesting path dependent betting strategies as well (such as the martingales), or some mixture of betting strategies.

(i) Fixed Probability Bettors

The most common and popular “strategy” found in this population is the fixed probability betting strategy. In this strategy, the gambler chooses some probability, usually $p > 0.5$ and continually bets. In evaluating gamblers who bet with fixed probabilities or wagers, we can observe what happens when a gambler has no strategic time inconsistency (irrespective to exit strategies). In a very heuristic way, we can label these kinds of gamblers as more risk seeking if $p < 0.5$ and less risk seeking if $p > 0.5$.

An example of this strategy is found in the betting history of the individual label as Professional #5619 in our dataset (see Fig. 12). Through the three date periods of 11/23/2017 3:17:43 PM - 11/23/2017 7:54:43 PM, 11/24/2017 8:22:40 AM - 11/24/2017 11:30:14 PM, and 11/25/2017 8:20:38 AM - 11/25/2017 10:04:12 PM, this gambler gambled 147 (whale) times, all at $p = 0.49$. This is an example of a simple, path-independent strategy. No matter what his results are, the gambler sticks to his initial chosen betting strategy. Upon entering the casino, the gambler commits to this strategy, and even upon accumulating losses, this individual continues to gamble at a suboptimal probability. In gambling at $p < 0.5$, this gambler is taking significant risk - in the long run the expectation runs negative rapidly.

Figure 12: Examples of gamblers using a fixed probability strategy. Shown are (a), (b), (c) the wager sizes, (d), (e), (f) cumulative profits, and (g), (h), (i) scaled cumulative profits over a length of consecutive gambings, corresponding to three occasions, respectively. This gambler consistently fixed the probability of winning to be $p = 0.49$, but exhibited inconsistent exit strategies (when to stop gambling during the day).

This gambler may have a time inconsistency in terms of exit strategies (cf. Figs. 12d, 12e, and 12f), but this is something we cannot empirically deduce. This individuals follows the betting

strategy, but exits once the losses reach some arbitrary stop-loss. We also observe loss-chasing behavior in the third day (Fig. 12f). This gambler begins by betting at an extremely large initial wager size of around 2 Ether (which is around the same size as the maximum bet he or she bet at in the past two days), and progressively increases bet sizing as losses accumulate (Fig. 12c). We observe a two-peak plateau in the bet sizing of this gambler. The first plateau is positive for the gambler. The gambler won two consecutive large bets (20 ETH), recovering a significant portion of their accumulated losses. Immediately, we observe a drastic decreasing in bet size, and a subsequent loss. The gambler continues to bet at high values, putting themselves more and more negative. After a string of consecutive losses, the gambler adjusts their bet size, eventually exiting a loss of 41.44 Ether (Fig. 12f). Attempting to chase a few initial losses through increasing bet sizing only resulted in a larger loss. It seems that this gambler followed a gain exit pattern (Fig. 12d). Upon large wins at the end of their first sessions, the gambler stopped betting (Fig. 12e).

(ii) Fixed Wager Bettors

Rarer is the fixed wager strategy as seen in our dataset. This strategy is extremely simple. A gambler takes some initial stake W_0 , and continually gambles, tuning his probability through wins and losses. Oddly, this strategy is never the only strategy the gambler employed over the timescale of a day.

(iii) Martingale Bettors

As the Etheroll minimum bet size is quite large, we observe that a martingale system diverges very quickly. However, certain gamblers still follow this system. As observed, this gambler follows a martingale strategy. Their bet size starts with an initial staking size $w_0 = 0.2$ Ether, and follows:

$$w_{i+1} = \begin{cases} 2^i w_0 & \text{if loss} \\ w_0 & \text{if win} \end{cases}$$

Where w_{i+1} is the $i + 1$ -th bet conditioned on i losses. In Fig. 13, we observe a very clear return to the initial staking size after every win. These graphs showcase the nearly linear staking gains of the martingale for a gambler from the timespan of 5/6/17 – 5/7/17 (where this individual betted a total of 172 times). This is because of the guaranteed return of w_0 from this system.

However, we notice on 5/6/17 there is a distinct drop in cumulative profits due the gambler having a stop loss after 6 consecutive lost bets (Fig. 13c). This showcases the dangerous fast divergence of the martingale system. We also see an interesting time inconsistency from this gambler. Looking at the data, we see the gambler's exact deviation from this strategy at their 69-th bet, where the gambler bets $2^3(0.2) = 1.6$ Ether at probability 0.5 and loses. If this gambler is following the martingale system, they should bet $2^4(0.2) = 3.2$ Ether at the same probability (to get a 1 : 1 return). However, the gambler disregards this, and gambles the same gamble of 1.6 Ether, and loses again. This causes a sharp decrease in their cumulative gains. Interestingly, the gambler makes a third bet of the same amount, and same probability. Probabilistically, this gambler will win this bet on average, but it comes with significant risk. This showcases the tendency of gamblers to chase losses - similar to the idea of gain-exit strategies. It also matches with the concept of the value function (according to the cumulative prospect theory [13], see our detailed discussion below). Theoretically, when making decisions under risk, gamblers become risk seeking when faced with losses. This helps to explain the "loss-chasing" phenomenon.

Other possible reasons for this sudden time inconsistency in strategy could involve the gambler's total wealth (wallet size), overall bankroll. From observing their wager sizing, we observe that the gambler never bets past 1.6 Ether. Hypothetically, the next bet in the martingale sequence (3.2) could simply be too much for the gambler to continue, forcing the gambler to deviate from the planned strategy.

The trajectory of the scaled cumulative profit of this gambler shows their valuation ("satisfaction") through the losses and gains, framed near the exit time (Fig. 13d, 13e). We observe

Figure 13: Example of martingale gamblers. Shown are (a), (b) the wager sizes, (c), (d) cumulative profits, and (e), (f) scaled cumulative profits over a varied length of consecutive gambings, corresponding to two occasions, respectively. Martingale strategy imposes significant risk, yet can be successful.

that even though this gambler was a net positive (on 5/6/17), their scaled cumulative profit is very faintly positive, representing the effect of the severe loss. In contrast, we see the effect of ending with a large win on 5/7/17. The gambler’s scaled cumulative profit is at a global maximum at their exit time, influencing their exit time. This gambler ended up as a net winner, winning 4.34 Ether.

3. Discussion and Conclusion

One of the most well-known models for evaluating how people behave under risk situations is the cumulative prospect theory model [13]. This model is an advancement on their original prospect theory model [16], which was based on common characteristics in decision makings,

including the framing effect, nonlinear preferences, source dependence, risk seeking behavior, and loss aversion. The “framing effect” is the idea that humans make decisions relative to a reference point, rather than the actual result. People making decisions under risk also often exhibit a tendency towards risk seeking behavior, such as preferences towards low probability tail events and preferring substantial probabilities of a larger loss over sure losses (see Figs. 1,11 and Tabs. 2,3). In tandem, loss aversion is exhibited in experiments with losses and gains. Individuals in general were found to be more affected by losses and gains, rather than final cumulative profit levels. Based on prior studies, Tversky and Kahneman noticed a distinct asymmetry in the preferences of gamblers towards gains over losses, too significant to be attributed to risk aversion or income effects [16]. Moreover, another important theoretical framework applicable to our work is the Barberis’ casino model [15], which assumes gamblers with cumulative prospect theory (CPT) preferences in the context of a casino. As Etheroll is a dice game, the Barberis’ casino model provides insights into understanding gamblers’ choices of “gain-exit” vs “loss-exit”. In Ref. [17], the authors investigate Barberis’ casino model with a focus on allowing randomized and path-dependent strategies. As detailed in the Results section, we explore gambling behavior and risk attitudes through theoretical foundations laid by these prior works.

Through our data analysis, we discover interesting patterns of gambling behavior. One of these findings is that as expected, a large proportion (approximately 60%) of gamblers are recreational (with 0-10 total lifetime bets). Additionally, we observe a nearly normal distribution ($\mu < 0$) of gamblers by their cumulative profit, which is expected from Etheroll’s 1% house commission. Surprisingly, gamblers on Etheroll also follow a generally more loss-averse distribution of probabilities ($p > 0.5$). We also find some interesting gamblers who follow betting strategies (path-independent and path-dependent), and attempt to qualitatively explain their exit times, behavior and risk profiles through the lens of prospect theory and cumulative profit. In looking at a mixture of strategies, we see gamblers who display strategic time inconsistencies, gain-exit and loss-exit strategies, and loss-chasing behavior (for example, see, Figs. 12, and 13).

To account for the individual heterogeneity in their utility function, it would be of interest to empirically approximate the cumulative prospect theory’s value function [13]. The barrier of applying this method directly to the current dataset comes from the fact that many gamblers do not necessarily bet amongst the whole spectrum of probabilities (0, 1), but instead bet within some chosen subset. Thus, it becomes almost impossible to estimate the whole distortion function. Having an estimate of the probability distortion or value function would allow us further to quantify the risk attitudes of individuals. It would be interesting to explore other publicly available gambling datasets to see if certain gamblers have weighting functions which underweight low probabilities, but overweight high probabilities. Individuals with value functions that have steeper loss regions are more sensitive to losses, and thus more willing to take risks and chase losses. Our present study paves the way for, and will ~~sit~~stimulate, future work in this direction.

We note that it may be promising for future work to take advantage of machine learning algorithms to search for other typical path-dependent strategies. Many other negative-progression, staking strategies exist, such as the D’alambert, Fibonacci, and Labouchere systems. However, it is very challenging to classify gamblers as gambling under these strategies, as it is quite rare for a gambler to perfectly follow a prescribed strategy. Being able to observe the deviation of gamblers from their initial strategies would pave the way to characterize the risk attitudes of the gamblers, and possible gain-exit vs. loss-exit patterns. Of particular interest is to find the existence of gamblers who are clearly loss-exiting against gain-exiting. Besides, understanding individual gambling behavior of choosing between and deploying betting systems that have varying risks from the perspective of the multi-armed bandit problem [18] is both interesting and promising.

4. Dataset and Methods

Dataset. This paper entirely focuses on data collected from bets on the DApp (Decentralized Application) Etheroll. Data about these bets used to be hosted on a site <https://www.cryptocurrencychart.com/etheroll-live-stats>. To obtain this data, we used a customized screen scraper built in Python using Requests and BeautifulSoup. Requests is a user-friendly Python library designed to handle HTTP requests, and BeautifulSoup is a HTML parsing library. We then databased this data, which consisted of approximately 250,000 individual bets and 2600 gamblers in a MySQL database. These numbers are based off the four iterations of Etheroll’s smart contract. Contract 1 ranges from 4/17/2017 to 4/24/2017, Contract 2 ranges from 5/4/2017 - 5/18/2017, Contract 3 ranges from 5/23/2017 - 10/25/2017, and Contract 4 ranges from 10/25/2017-12/12/2017. All of this data was collected when the minimum bet on Etheroll was still 0.1 Ether (a value which ranged roughly from 4.3 USD to 52 USD).

We present simple descriptive statistics for each contract as given in Tab. 4.

Table 4: Environment Summary Statistics

Contract	Average Bet Size (ETH)	Unique Gamblers	Total Bets
1	1.595	201	3919
2	2.006	179	5671
3	1.527	1889	132826
4	0.987	516	43425

In our dataset, a raw individual bet consists of 7 fields: A Date timestamp, Player identification (such as Professional #2924, All in #2922), Bet Size (in Ether), Chance (number chosen to roll under, referred to as the probability of winning in the main text), Paid_ETH (Payout in Ether), and Paid_USD (Payout in USD, converted from Ether at the time). According to the maker of the website, the player identification names follow this pattern: Newbie - New address, All in - High value bets, Lucky - Wins against the odds, Play it safe - Multiple high chance bets, Against the odds - Losses with high win chance, One in a million - Won very low chance bet, Intermediate - more than 5 bets, Professional - more than 25 bets, Legend - more than 100 bets.

The Chance feature is useful for determining the general riskiness of the population. Additionally, having individual player identification codes allows us to subset our data to observe the complete gambling history of each individual gambler. Having access to the timescales of the gamblers also lets us observe some interesting time inconsistencies prevalent in the data. Lastly, knowing the Bet Sizes and Payouts per bet allows us to solve for the cumulative profit of the gambler at each time step. This allows us to understand our results through the lenses of the cumulative prospect theory [13] and the Barberis’ casino model [15].

We subset and clean up our dataset by pre-processing the raw data. The first step is to subset the dataset in order to organize the data by individual gamblers. Next, we take these individual gamblers and subset them by the days they gambled in. Lastly, we compute the simple payout W for each gamble, which for some bet B , result $R \in \{\text{loss}, \text{win}\}$ was:

$$W = \begin{cases} 0, & R = \text{loss} \\ W(1 - p - e)/p, & R = \text{win} \end{cases}$$

Where e is the house commission, 0.01, and p is the probability of winning the gamble.

Methods. We explore this dataset using a variety of common data analysis techniques. We collect, analyze and visualize data using the Python’s pandas library. As mentioned above, the data was scraped from <https://www.cryptocurrencychart.com/etheroll-live-stats>. An alternative way to obtain this dataset would involve setting up a computer as a Geth node

(Ethereum node member), and use the JSON-RPC API to repeatedly call for contract log events (where the smart contract specifies data).

First, we aim to characterize gamblers in Etheroll, through each of their lifetime bet, bet size, betting probability, and cumulative profit distributions. We will determine patterns in this data by analyzing histogram distributions. This will allow us to profile the average gambler's preference for probabilities of winning, and average bet frequency. Additionally, we will analyze the relationship between bet sizing and probabilities of winning.

Next, we aim to carefully characterize gamblers who follow a few prescribed types of gambling strategies. We first preselect a few path-independent (fixing of probabilities and wager size) and path-dependent strategies (martingale). To explain certain psychological behavior, such as loss function, we will explain the behavior in terms of prospect theory concepts such as the weighting function and value function. Additionally, we will try to qualitatively define the risk profile of gamblers, depending on strategies and gambling patterns. We do this through observing cumulative profits over time and changes in wager sizing. To characterize the local behavior near a gambler's exit time, we use the metric of scaled cumulative profit (Figs. 12h-j and 13d-e). Take some set of times $T = \{1, \dots, \tau\} \subset \mathbb{N}$, where τ is a forced exit time, and set of cumulative profits $W = \{w_0, w_1, \dots, w_\tau\}$, where for some $t \in T$, $w_t = \sum_{i=0}^t r_i$, where r_i is the profit at time i . We scale this cumulative profit, instead taking an alternate sequence $\{2^{-i} w_i\}_{i=0}^\tau$. This is an attempt to model the memory of a gambler, with the assumption that more recent bets are weighted more significantly than bets in the past. This follows a psychological phenomena known as the primacy and recency effect, where initial and final results are overweighted in terms of importance and memory [19].

Ethics. Ethical assessment is not required prior to conducting the research reported in this paper, as the present study does not have experiments on human subjects and animals, and does not contain any sensitive and private information.

Data Accessibility. Data and relevant code for this research work are stored in GitHub: <https://github.com/Orangead/Understanding-Risk-Attitudes> and have been archived within the Zenodo repository (DOI: 10.5281/zenodo.4044242): <https://doi.org/10.5281/zenodo.4044242>.

Authors' Contributions. J.M. and F.F. conceived the project; J.M. collected the data, performed data analyses, plotted the figures, and wrote the first draft of the manuscript; F.F. analyzed the data and contributed to revisions of the manuscript draft. All authors gave final approval for publication.

Competing Interests. We have no competing interests to declare.

Funding. J.M. is grateful for the support by the Neukom Scholars Program. F.F. thanks the generous support by the Neukom Faculty CompX grant and the VeChain-Neukom Blockchain Research grant.

Acknowledgements. The authors would like to thank Herbert Chang and James Detweiler for helpful discussions.

Disclaimer. We have no disclaimer to disclose.

References

1. SM R, D Y, Lee S. 1995 Client/Server Computing Technology: A Framework for Feasibility Analysis and Implementation. *International Journal of Information Management* **15**, 135–150.
2. What is Torrenting? 4 Things You Need to Know. <https://www.cloudwards.net/what-is-torrenting/>.
3. Buterin V Ethereum Whitepaper: A NEXT GENERATION SMART CONTRACT AND DECENTRALIZED APPLICATION PLATFORM. .
4. Nakamoto S Bitcoin Whitepaper. <https://bitcoin.org/bitcoin.pdf>.
5. David Nick J. 2015 Data-Driven De-Anonymization in Bitcoin. <https://nickler.ninja/papers/thesis.pdf>.
6. Bracci A, Nadini M, Aliapoulios M, McCoy D, Gray I, Teytelboym A, Gallo A, Baronchelli A. 2020 The COVID-19 online shadow economy. *arXiv preprint arXiv:2008.01585*.
7. De Domenico M, Baronchelli A. 2019 The fragility of decentralised trustless socio-technical systems. *EPJ Data Science* **8**, 2.
8. ElBahrawy AY, Alessandretti L, Baronchelli A. 2019 Wikipedia and digital currencies: interplay between collective attention and market performance. *Frontiers in Blockchain* **2**, 12.
9. Li TR, Chamrajnagar A, Fong X, Rizik N, Fu F. 2019 Sentiment-based prediction of alternative cryptocurrency price fluctuations using gradient boosting tree model. *Frontiers in Physics* **7**, 98.
10. Wang X, Pleimling M. 2019 Online gambling of pure chance: Wager distribution, risk attitude, and anomalous diffusion. *Scientific Reports* **9**, 1–17.
11. State of the DApps: 1,539 Projects Built on Ethereum. <https://www.stateofthedapps.com/>.
12. Etheroll : Smart Contract. <https://etheroll.com/#/smart-contract>.
13. Tversky A, Kahneman D. 1992 Advances in Prospect Theory: Cumulative Representation of Uncertainty. *Journal of Risk and Uncertainty* **5**, 297–323.
14. Ethereum (ETH) Historical Data | CoinMarketCap. <https://coinmarketcap.com/currencies/ethereum/historical-data/>.
15. Barberis N. 2011 A Model of Casino Gambling. *Management Science* **58**, 35 – 51.
16. Tversky A, Kahneman D. 1979 Prospect Theory: An Analysis of Decision under Risk. *Econometrica* **47**, 263–291.
17. ~~Morrison AB, Conway AR, Chein J. 2014 Primacy and recency effects as indices of the focus of attention. *Front Hum Neurosci* **8**.
Ethash.~~
17. He XD, Hu D, Obloj J, Zhou XY. 2017 Technical Note – Path-Dependent and Randomized Strategies in Barberis’ Casino Gambling Model. *Operations Research* **65**, 97–103.
18. Huo X, Fu F. 2017 Risk-aware multi-armed bandit problem with application to portfolio selection. *Royal Society open science* **4**, 171377.
19. Morrison AB, Conway AR, Chein J. 2014 Primacy and recency effects as indices of the focus of attention. *Front Hum Neurosci* **8**.
20. [Ethash](https://github.com/ethereum/wiki/wiki/Ethash). <https://github.com/ethereum/wiki/wiki/Ethash>.